



# Year-round records of bulk and size-segregated aerosol composition in central Antarctica (Concordia site) Part 1: Fractionation of sea-salt particles

Michel Legrand[1,2], Susanne Preunkert[1,2], Eric Wolff[3], Rolf Weller[4], Bruno Jourdain[1,2], and Dietmar Wagenbach[5*]

[1]Université Grenoble Alpes, CNRS, Laboratoire de Glaciologie et Géophysique de l'Environnement (LGGE), Grenoble, 38402, France
[2]CNRS, Laboratoire de Glaciologie et Géophysique de l'Environnement (LGGE), Grenoble, 38402, France
[3] Department of Earth Sciences, University of Cambridge, Cambridge, CB2 3EQ, UK
[4]Alfred Wegener Institut für Polar und Meeresforschung, Bremerhaven, 27570, Germany
[5]Institut für Umweltphysik, University of Heidelberg, Heidelberg, 69120, Germany
[*] Deceased December 2014

Correspondance to M. Legrand (Michel.Legrand@univ-grenoble-alpes.fr)

**Abstract.** Multiple year-round records of bulk and size-segregated composition of aerosol were obtained at the inland site of Concordia located at Dome C in East Antarctica. In parallel, sampling of acidic gases on denuder tubes was carried out to quantify the concentrations of HCl and $HNO_3$ present in the gas phase. These time-series are used to examine aerosol present over central Antarctica in terms of chloride depletion relative to sodium with respect to freshly emitted sea-salt aerosol as well as depletion of sulfate relative to sodium with respect to the composition of seawater. A depletion of chloride relative to sodium is observed over most of the year, reaching a maximum of ~20 ng m$^{-3}$ in spring when there are still large sea-salt amounts and acidic components start to recover. The role of acidic sulfur aerosol and nitric acid in replacing chloride from sea-salt particles is here discussed. HCl is found to be around twice more abundant than the amount of chloride lost by sea-salt aerosol, suggesting that either HCl is more efficiently transported to Concordia than sea-salt aerosol or reemission from the snow pack over the Antarctic plateau represents an additional significant HCl source. The size-segregated composition of aerosol collected in winter (from 2006 to 2011) indicates a mean sulfate to sodium ratio of sea-salt aerosol present over central Antarctica of 0.16 ± 0.05, suggesting that, on average, the sea-ice and open ocean emissions equally contribute to sea-salt aerosol load of the inland Antarctic atmosphere. The temporal variability of the sulfate depletion relative to sodium was examined at the light of air mass backward trajectories, showing an overall decreasing trend of the ratio (i.e. a stronger sulfate depletion relative to sodium) when air masses arriving at Dome C had travelled a longer time over sea-ice than over open-ocean. The findings are shown to be useful to discuss sea-salt ice records extracted at deep drilling sites located inland Antarctica.





Keywords: Sea-salt aerosol, hydrochloric acid, sea-salt fractionation, chemistry (chemical composition and reactions)

## 1. Introduction

The understanding of the atmospheric budget of sea-salt aerosol at high latitudes is important for several reasons. In these regions, sea-ice-related processes such as frost flowers (Wagenbach et al., 1998; Rankin et al., 2000) or blowing snow (Yang

et al., 2008; Jones et al., 2009) could represent an important sea-salt aerosol source with respect to the common sea-salt emissions from open-ocean. If correct, that offers the possibility to reconstruct the sea-ice conditions in the past by studying sea-salt ice core records (Rankin et al., 2002). Sea-salt aerosol emitted from open-ocean and by sea-ice related processes represents a large atmospheric source of halogens (Sander et al., 2003) that, if activated, can contribute to the chemical reactivity of the atmosphere over these high latitude regions (see Simpson et al. (2007) and Abbatt et al. (2012) for reviews).

The chemical composition of sea-salt aerosol varies, depending on the emission process involved. In particular, the sulfate to sodium mass ratio (R) of sea-salt aerosol emitted by the open ocean is similar to that in seawater (0.25) whereas sea-ice-related emissions lead to a strong depletion of sulfate relative to sodium (R well below 0.25) in aerosol, due to precipitation of mirabilite ($Na_2SO_4.10\ H_2O$) during freezing of seawater (Wagenbach et al., 1998). At coastal Antarctic sites in winter, where the sea-ice surface was shown to be the dominant source of sea-salt aerosol, Wagenbach et al. (1998) reported R

values close to 0.07 at Neumayer (NM, 70°S 85°W) and 0.10 at Dumont d'Urville (DDU, 66°S 140°E). These estimations were done by examining the relationship between the non-sea-salt sulfate ($nssSO_4^{2-}$) content calculated by using the seawater ratio of sulfate to sodium ($nssSO_4^{2-} = SO_4^{2-} - 0.25\ Na^+$) and sodium in bulk aerosol samples. A study of the size-segregated aerosol composition performed at DDU has shown that sea-salt aerosol present in super micron modes is depleted in sulfate relative to sodium (R close to 0.13) from May to October (Jourdain and Legrand, 2002). This more direct quantification of

the sulfate depletion relative to sodium in aerosol with respect to seawater confirmed the importance of sea-ice related processes as a source of sea-salt in winter at the Antarctic coast.

At inland Antarctic sites, the estimation of the sulfate depletion relative to sodium by direct examination of the bulk aerosol composition becomes far more difficult due to low sea-salt concentrations and high biogenic sulfate content, as discussed by Weller and Wagenbach (2007) for Kohnen (75°S, 0°E) and Jourdain et al. (2008) for Concordia. At Concordia, the size-

segregated aerosol composition was studied over the course of winter 2006 using a 12-stage impactor (Jourdain et al., 2008) and over the years 2005-2007 using a 4-stage impactor (Udisti et al., 2012). These two studies identified significant sulfate depletion relative to sodium during a few winter marine events. They also pointed out that, even when examining the chemical composition of particles deposited on the top stages of the impactor, there can be a significant underestimation of the degree of fractionation of sea-salt particles due to a residual presence of biogenic sulfate.

In addition to the sea-salt fractionation discussed above, related to the presence of sea-ice in winter, the more commonly observed release of chloride from sea-salt particles (Graedel and Keene, 1995) is also taking place in summer in coastal Antarctica. This is suggested by values of the chloride to sodium ratios (r) in bulk aerosol filters that are lower than the





seawater reference value of 1.8 (Wagenbach et al., 1998, Jourdain and Legrand, 2002; Legrand et al., 2016). For inland Antarctica, bulk aerosol composition also revealed r values lower than 1.8 in summer (Hara et al., 2004; Cunningham and Zoller, 1981; Tuncel et al., 1989). It remains however difficult to quantify how much of the chloride loss had occurred in the atmosphere or on sea-salt accumulated on the bulk filter (an acid-induced sampling artefact). More reliable quantification of

the chloride loss can be made by sampling aerosol using an impactor for which interactions between sea-salt particles and gaseous acids or acidic aerosols are far more limited than on a bulk aerosol filter. In this way, using a 12-stage impactor, Jourdain and Legrand (2002) and Kerminen et al. (2000) reported chloride loss in summer at the coastal Antarctic sites of DDU and Aboa, respectively. The few impactor data available for inland Antarctica (Jourdain et al., 2008; Udisti et al., 2012) have been discussed with respect to sulfate to sodium depletion in winter but not with respect to the chloride loss.

The release of chloride from sea-salt is in the form of HCl for which atmospheric measurements are very rare in Antarctica. Using denuder tube sampling, a few measurements were done at the coastal site of DDU (Jourdain and Legrand, 2002). However, as discussed by Barrie et al. (1994), the denuder tube sampling of HCl in the marine boundary layer, where a large amount of particulate chloride is present, remains difficult and the reliability of such data needs to be carefully examined. Hara et al. (2004) reported HCl data obtained by deploying at the coastal site of Syowa a sampling line made by a PTFE

filter and alkaline impregnated filters for which a sampling artefact is also possible. Finally, the examination of the ionic composition (anions versus cations) of South Polar snow layers clearly suggested the presence of HCl in the atmosphere in summer at inland Antarctic sites (Legrand, 1987).

Here we report on year-round atmospheric observations of sea-salt aerosol done at the Concordia site located on the high east Antarctic plateau since 2006. The 10-year record (2006-2015) of bulk aerosol is complemented by a study of the size-

segregated aerosol composition conducted by running a 12-stage impactor. Initiated in 2006, the impactor samplings were done continuously over three years (2009-2011). Impactor data are used in this paper for accurately quantifying the contribution of sea-ice related emissions to the sea-salt budget over inland Antarctica and are also essential for discussing biogenic sulfur aerosol as done in the accompanying paper from Legrand et al. (this issue). Gaseous acidic species were also sampled year-round since 2009 at Concordia using coated denuder tubes. Results for HCl are of interest here to discuss the

atmospheric budget of chloride in these remote regions. Some of the findings are also discussed with respect to interpretation of deep ice core chemical records.

## 2. Sites, Sampling, and Analyses

As summarized in Fig. 1, samplings of various gases and atmospheric aerosol were initiated in 2006 at the inland site of Concordia (75°06'S, 123°20'E, 3233 m asl, located 1100 km away from the nearest coast of East Antarctica). Aerosol was

sampled at a flow rate of 0.834 $m^3$ STP (standard temperature and pressure conditions of 298 K and 1013 hPa) $min^{-1}$ on circular quartz filters (Gelman Pallflex Tissuquartz 2500QAT-UP, 15 cm diameter), denoted as HV (high volume) in Fig. 1. Forty one filters were collected in 2006 and 405 filters from January 2008 to January 2016. In 2006, each weekly sampling





was interrupted over one or two days whereas since 2008 the weekly sampling was most of the time conducted continuously (Fig.1). A piece of each filter (10 cm$^2$ of a total surface of 150 cm$^2$) was extracted with 10 mL of ultra pure Milli-Q water. Twelve times per year a field blank was achieved. Given the weekly sampling time, a large air volume was sampled (~ 8000 m$^3$) permitting the blank values to remain well below 1 ng m$^{-3}$ (0.17 ± 0.25 ng m$^{-3}$ for chloride, 0.17 ± 0.15 ng m$^{-3}$ for

sodium, 0.4 ± 0.3 ng m$^{-3}$ for sulfate, 0.05 ± 0.05 ng m$^{-3}$ for nitrate, and zero for MSA). Note that due to an error made in the field, different filters were used in 2007 leading to high sodium blank values. These data were therefore not considered in this paper (Fig. 2).

In addition to bulk HV aerosol samplings, a multiple year-round study of size-segregated aerosol composition has been carried out at Concordia using 105 discontinuous samplings done between March 2006 and January 2012 using a small

deposit area impactor, similar to the one developed by Maenhaut et al. (1996), and equipped with a 20 µm cut-off diameter inlet. At each run of the impactor a blank of the deposit was done. A sampling interval of 2 weeks was applied with a flow rate of 0.54 m$^3$ h$^{-1}$. In 2006 and 2007 eight run per year were done, whereas a more continuously sampling (25 runs per year) was done from 2009 to 2012 (Fig. 1). With a sampled air volume of 160 m$^3$ and using a 9 mL extraction water volume, the blank values of the deposit remain well below 1 ng m$^{-3}$ (0.22 ± 0.18 ng m$^{-3}$ for chloride, 0.17 ± 0.12 ng m$^{-3}$ for sodium, 0.08

± 0.06 ng m$^{-3}$ for sulfate, 0.06 ± 0.04 ng m$^{-3}$ for nitrate, and zero for MSA). All data were blank corrected.

Finally, 261 samplings of acidic gases including HCl and HNO$_3$ were done at Concordia between January 2009 and March 2014 using a gas sampling line of 3 annular denuder tubes placed in series and coated with Na$_2$CO$_3$ (Jourdain and Legrand, 2002; Legrand et al., 2012). Over the first year, a sampling interval of 2 to 3 days was applied in summer and winter, respectively. From 2010 to 2016 the sampling became more continuous (Fig. 1) and the sampling time was increased to 6

days both for winter and summer. At each denuder tube run, two tubes were used as blanks to evaluate the contamination during coating and drying of the tubes. With a sampling interval of 6 days and a flow rate of 0.6 m$^3$ hr$^{-1}$ (i.e. a total air sampled volume of ~ 80 m$^3$) and extraction of tubes with 5 mL of ultra pure Milli-Q water, the blank values correspond to atmospheric concentration in the range of 0.1 to 0.4 ng m$^{-3}$ for nitrate. For chloride, the blank values were far more significant (up to 30 ng m$^{-3}$ in 2009) and showed a large variability (± 10 ng m$^{-3}$). These samplings were discarded from the

record and, as seen in Fig. 3, there are far more nitrate than chloride data in 2009. The situation improved when a more continuous sampling was applied and since mid-2012 chloride blank values remained most of the time limited to less than 1 ng m$^{-3}$ (reaching occasionally 1 to 4 ng m$^{-3}$). These blank values were subtracted from raw data.

To characterize the origin of air masses reaching the Concordia region, 5- and 10-day backward trajectories were computed using the Hybrid Single-Particle Lagrangian Integrated Trajectory model (Stein et al. (2015), available at:

http://ready.arl.noaa.gov/HYSPLIT.php). Meteorological data from Global Data Assimilation Process (available at ftp://arlftp. arlhq.noaa.gov/pub/archives/gdas1) were used as input, and the model was run every 6 h in backward mode for three different altitudes (0, 250, and 500 m above ground level, agl).





## 3. Discussions

### 3.1. Chloride relative to sodium fractionation in aerosol

To evaluate the magnitude of the chloride to sodium fractionation of aerosol collected at Concordia with respect to the seawater composition, we calculate the chloride to sodium mass ratio (r). The uncertainties of r are related to the accuracy of

the determinations of sodium and chloride that are determined by the ion chromatography accuracy (5%) and the standard deviation of blank filter values ($\sigma_{blank}$), as follows:

$$\Delta r^2 = (\Delta Cl/Na)^2 + (Cl\ \Delta Na/Na^2)^2 \qquad (1)$$

with $\Delta Cl^2 = (0.05Cl)^2 + \sigma_{blank}^2$

and $\Delta Na^2 = (0.05Na)^2 + \sigma_{blank}^2$

As discussed in Sect. 2, HV blank values lead to a $\sigma_{blank}$ of 0.15 ng m$^{-3}$ for sodium and 0.25 ng m$^{-3}$ for chloride.

We also calculated the depletion of chloride relative to sodium for aerosol (Cl$_{depletion}$) with respect to the composition of fresh sea-salt aerosol as follows:

$$Cl_{depletion} = k_{Cl/Na}\,(Na^+) - (Cl^-) \qquad (2)$$

With $k_{Cl/Na}$ being the ratio in fresh sea-salt aerosol.

In summer, as discussed by Legrand et al. (2016), sea-salt aerosol reaching central Antarctica essentially originates from the open-ocean and in applying equation (2) we have assumed a $k_{Cl/Na}$ value of 1.8 (i.e. the seawater value). The uncertainties in calculating the chloride depletion relative to sodium are calculated as:

$$\Delta(Cl_{depletion})^2 = (1.8\ \Delta Na)^2 + (\Delta Cl)^2 \qquad (3)$$

In winter, when sea-ice related processes act as sources of sea-salt aerosol, the precipitation of mirabilite on the sea-ice

surface which causes a loss of sulfate relative to sodium, also causes a loss of sodium relative to chloride, leading to a $k_{Cl/Na}$ value slightly higher than 1.8 (Wagenbach et al., 1998). A mass balance calculation, done by assuming that the totality of sulfate (i.e. 0.25 Na) has been removed by mirabilite precipitation, permits estimation of an upper limit of the subsequent enrichment of chloride relative to sodium with a $k_{Cl/Na}$ value reaching 2.2 (Legrand et al., 2016). As discussed in sect. 3.2, from May to October a variable degree of sulfate depletion relative to sodium was observed on sea-salt particles present at

Concordia. We therefore have assumed that over this period the $k_{Cl/Na}$ value can range between 1.8 and 2.2 ($k_{Cl/Na} = 2.0 \pm 0.2$). In winter, the uncertainties in calculating the chloride depletion relative to sodium therefore include uncertainties in the $k_{Cl/Na}$ value, as follows:

$$\Delta(Cl_{depletion})^2 = (k_{Cl/Na}\ \Delta Na)^2 + (Na\ \Delta k_{Cl/Na})^2 + (\Delta Cl)^2 \qquad (4)$$

With $k_{Cl/Na}$ being equal to 2.0 and $\Delta k_{Cl/Na}$ equal to 0.2.

The r values in bulk aerosol show a large departure from 1.8 in summer with values remaining far lower than 1.8 (close to 0.1 from November to April, Fig. 2). In winter, monthly mean values reach a maximum of $1.2 \pm 0.3$ in July (not shown) but, as seen in Fig.2, a few weekly samples exhibit r values that significantly exceed 1.8. The grand average r value over the year is about 0.7.





The calculated chloride depletion relative to sodium on bulk aerosol has an annual mean of 8.4 ± 15 ng m$^{-3}$, remains close to 4 ± 1.6 ng m$^{-3}$ from January to September and increases in spring (21 ± 9 ng m$^{-3}$ in October, 26 ± 12 ng m$^{-3}$ in November, and 9 ± 2.6 ng m$^{-3}$ in December, Fig. 4a). We cannot however rule out that these values are overestimated due to an acid-induced remobilization of chloride on the HV filters. The comparison of HV data with those available on the impactor (for which the above mentioned artefact is strongly limited) in 2009 and 2010 indicates a similar timing of the maximum of the chloride depletion (October-November, Fig. 5) but lower values than on the HV filters (18 ± 9 ng m$^{-3}$ against 36 ± 15 ng m$^{-3}$, not shown), suggesting that the chloride depletion calculated from HV data is overestimated. On an annual basis (2009-2010), the difference between HV and impactor samplings is relatively lower with a chloride depletion calculated on the HV filters of 10.4 ± 14 ng m$^{-3}$ against 7.4 ± 7 ng m$^{-3}$ on the impactor runs, likely due to the fact that the HV sampling artefact would be important only when acidic species are abundant in spring/summer. From that, we estimate that the overestimation of the chloride depletion calculated on the HV filter sampling is on an annual average close to 40% but reaches 100% in spring.

Several previous studies discussed the nature of chemical species involved in the de-chlorination of sea-salt aerosol in Antarctica but no overall picture yet emerged. For instance, based on 12-stage impactor data, Kerminen et al. (2000) found that in January 1998 at the coastal site of Aboa, H$_2$SO$_4$ particles account for slightly more of the observed de-chlorination than HNO$_3$ whereas Jourdain and Legrand (2002) attributed most of the de-chlorination to HNO$_3$ with little effect of the sulfur compounds at DDU (end of November to mid December 2000). Rankin and Wolff (2003) also showed that at coastal Halley station nitrate was reasonably well-correlated with sodium on the stages of an impactor associated with smaller sea-salt aerosol, suggesting an important role for nitrate in reacting with sea-salt there. Until now no impactor data were available to discuss species involved in the chloride loss over central Antarctica. The presence of nitrate on HV filters seen during later winter/spring (Fig. 4d) when nitric acid concentrations become large (Fig. 4f), suggests a significant role of nitric acid in acidifying sea-salt particles over the Antarctic plateau. That is clearly confirmed by the examination of the size-segregated composition of aerosol (Fig. 6) showing the presence of nitrate mainly on sea-salt particles (and to a far lesser extent on sulfuric acid submicron particles).

An examination of the role over time of chemical species possibly involved in aerosol de-chlorination is reported in Fig. 7, from end-winter to summer in 2009 and 2010. The amount of chloride loss is compared to the two main atmospheric acidic components, nitrate and acidic sulfur aerosols. We have calculated the acidic sulfur component as the sum of non-sea-salt sulfate plus MSA after subtracting the amount of ammonium. This examination was done for both small (0.09 to 0.46 µm diameter, Fig. 7a) and large (0.64 to 3.5 µm diameter, Fig. 7b) particles. We also report the temporal evolution of the observed chloride concentration (in grey) and of the chloride fraction that was lost (in blue) after emission. The sum of these two fractions corresponds to the amount of chloride that was originally present as sea-salt, as estimated from the observed sodium concentration and the relevant $k_{Cl/Na}$ value for freshly emitted sea-salt aerosol (see equation 2). Finally, we report the time evolution of nitric acid collected in denuder tubes (Fig. 7c). The first feature revealed by Fig. 7 is that the maximum of the chloride loss (in blue in Figs. 7a and 7b) takes place from mid/end October to mid/end November. That coincides with





the period at the end of winter when there are still important sea-salt concentrations reaching Concordia (see the sum of grey and blue fractions in Figs. 7a and 7b) before the decrease by late-November/early-December and the overall increase of both submicron sulfur particles (in green in Fig. 7a) and gaseous nitric acid (in red in Fig. 7c). Although this applies to the absolute amounts of chloride lost, the percentage of chloride that is lost (see blue versus the sum of blue and grey in Figs. 7a

and 7b) remains low (down to 20%) in September but generally exceeds 50% (up to 100%) in November and during the rest of summer. That clearly suggests that the amount of chloride release from sea-salt depends on the sea-salt load and of the availability of acidic species to replace chloride on sea-salt aerosol. Fig. 7 also indicates that, as reported in many previous studies (see Graedel and Keene (1995) and references therein), the fraction of chloride lost with respect to emission (i.e., the blue compared to the sum of blue and grey in Fig. 7) is larger on small particles (Fig. 7a) than on large particles (Fig. 7b).

Whatever the time period, acidic sulfur particles are always present in large enough amounts to replace chloride in small sea-salt particles (Fig. 7a). A quite different picture emerges for larger particles with a competition between gaseous nitric acid and acidic sulfur compounds (Fig. 7b). Only during full summer conditions is there enough acidic sulfur present on large particles, whereas in October/November there is insufficient sulfur and nitric acid becomes important in replacing chloride. Note that later in summer, when acidic sulfur species become abundant enough in large particles, they become the dominant

displacement agent rather than nitrate, despite the relatively high concentrations of nitric acid (Fig. 7c). This suggests that sea-salt aerosol is able to scavenge (mainly smaller) acidic sulfate particles, which then react to displace chloride. $HNO_3$ has less affinity for acidic particles, and might even itself be displaced from particles where it had already reacted.

As seen in Fig. 3, HCl exhibits typical concentrations remaining below 10 ng m$^{-3}$ in winter and a well-marked maximum in spring with values reaching 40 to 60 ng m$^{-3}$ in October and November. Given that the chloride depletion calculated for HV

filters is overestimated, its comparison with HCl concentrations measured on denuder tubes in 2013-2015 (Fig. 4g) suggests that, in spring and summer, HCl could be around a factor two more abundant than the amount of chloride lost by sea-salt aerosol (Fig. 4e and 4g). Several causes can be invoked to explain this difference. First, HCl may be more efficiently transported to Concordia than sea-salt aerosol, implying a longer atmospheric lifetime of HCl than sea-salt aerosol. Examination of HCl and sea-salt gradients between the coast and inland Antarctica tends to support this assumption. Sea-salt

aerosol decreases by more than one order of magnitude from the coast (200 ng m$^{-3}$ and 400 ng m$^{-3}$ of sodium at DDU in winter and summer respectively, Legrand et al., 2016) to Concordia (3 ng m$^{-3}$ and 15 ng m$^{-3}$ of sodium in summer and late winter, respectively, Figs. 4b and 4d) whereas HCl summer concentrations only decrease from the coast (~50 ng m$^{-3}$ at Syowa, Hara et al., 2004; ~100 ng m$^{-3}$ at DDU, Jourdain and Legrand, 2002) to 20-40 ng m$^{-3}$ at Concordia (Fig. 4g).

However, such a relatively weak gradient of HCl concentrations between the coast and inland Antarctica could also be

accounted for by an HCl reemission from the snowpack. The existence of HCl reemission from the snowpack was suggested by the observation of a trend of chloride levels in the upper meters of the snowpack, particularly at sites characterized by very low snow accumulating rates (less than 3 g H$_2$O cm$^{-2}$ yr$^{-1}$) like Concordia and Vostok (Legrand and Delmas, 1988; Legrand et al., 1996), whereas the phenomenon is strongly reduced at sites with higher snow accumulation rates (5 to 10 g cm$^{-2}$ yr$^{-1}$) like Dronning Maud Land (Weller et al., 2004) or the South Pole (Legrand et al., 1996). A more direct evidence of





the remobilization of HCl after its deposition in snow came from the observed presence of $^{36}$Cl in the surface snow at Vostok (Delmas et al., 2004), due to a broad peak starting in 1940 and ending nearby the surface instead of the expected peak in 1950-1960 related to atmospheric nuclear tests having taken place in the late 1950s to the early 1960s. This process could result from HCl formed in the atmosphere being deposited and re-emitted, allowing HCl effectively to hop inland through

several steps thus increasing its apparent lifetime. Alternatively it could also arise from reactions between acid and sea salt occurring in the snowpack itself, allowing new production of HCl that augments that occurring in the atmosphere. The fact that the grand average r value in aerosol at Concordia (0.7) is only slightly higher than the average (0.58, Röthlisberger et al., 2003) observed in snow between 10 m and 50 m depth (i.e., covering the last 1000 years) at Dome C (see Sect. 3.3) would suggest that post-depositional emissions may be largely of already produced HCl rather than new production in the

snowpack. Since, however, the r value of 0.7 is likely underestimated to due to an acid-induced remobilization of chloride on the HV filters, we cannot ruled out that a significant fraction of chloride is also lost due to HCl chemical production in the snow.

### 3.2. Sulfate relative to sodium fractionation in winter

In addition to the above-discussed depletion of chloride relative to sodium that mainly occurs in spring-summer, another

fractionation process that mainly takes place in winter is a sulfate depletion relative to sodium with respect to the seawater composition. We have quantified this process by examining the sulfate to sodium (R) ratio in sea-salt aerosol collected on the 12 stage impactor. At the coast, as done by Jourdain and Legrand (2002), R values can be directly derived from the examination of the sulfate and sodium present in submicron particles (diameter particles larger than 1.7 μm). However, at Concordia the separation of the biogenic and sea-salt sulfate modes is less straightforward than at coastal sites. As seen in

Fig. 6, there is still a small contribution of biogenic sulfate on particles having a diameter larger than 0.46 μm where the main fraction of sea-salt aerosol is present, and conversely, we can note the presence of fine sea-salt aerosol.

To evaluate the sulfate depletion relative to sodium of sea-salt aerosol reaching Concordia in winter, we therefore have corrected the concentration of sulfate present on the upper stages of the impactor for the biogenic contribution. This biogenic sulfate fraction can be estimated from the MSA concentrations and using the observed relationship between sulfate and MSA

on small particles (diameter of less than 0.46 μm). The effect of the small amount of sodium present on small particles on this relationship was examined by subtracting from sulfate the sea-salt fraction. This is done by assuming a sulfate to sodium ratio of sea-salt aerosol ranging between 0.25 (if no fractionation with respect to the seawater composition is assumed) and 0.07 (if the sulfate depletion relative to sodium is very large, as seen at the coast). Figs. 8 and 9 indicate that uncertainties related to the assumed value of sulfate to sodium ratio of sea-salt aerosol have only a weak effect on the derived slope

(denoted $k_{nssSO4/MSA}$) of the linear relationship between sulfate and MSA on the small particles (see the vertical bars reported in Figs. 8e and 9e). It is here important to emphasize that the $k_{nssSO4/MSA}$ values are systematically changing over the course of winter, showing higher values (16 ± 4.4, see Fig. 9 for instance) at mid-winter (June/-August) and lower values (8 ± 2.4, see Fig. 8 for instance) in late winter (September-October). The corresponding increase of the MSA to non-sea-salt ratio





(from 6% to 13%), already seen at the coast by Legrand and Pasteur (1998), is discussed in Legrand et al. (this issue) for Concordia in terms of source regions of biogenic sulfur aerosol over the course of the year.

For the 5 lowest stages (smallest particle sizes) of the impactor, we find that the ratio of $nssSO_4/MSA$ is reasonably constant (red dots in Fig 8e (or Fig 9e) falling on a straight line through zero). This then supports the assumption that in winter the

size distributions of biogenic sulfate and MSA are the same and that the $nssSO_4/MSA$ ratio is constant over the entire size distribution (see also Legrand et al., this issue). This allows us, for each impactor run, to calculate the sea-salt sulfate ($ssSO_4$) present on larger particles as follows:

$$ssSO_4 = SO_4 - k_{nssSO4/MSA} \, MSA \quad (5)$$

$k_{nssSO4/MSA}$ being the slope of the linear relationship observed between $nssSO_4^{2-}$ and MSA on the 5 lower stages of the

impactor, where most of biogenic sulfate is present.

The uncertainties in calculating $ssSO_4$ are related to the accuracy of the determinations of sulfate and MSA as well as the uncertainties of the calculated value of $k_{nssSO4/MSA}$:

$$\Delta(ssSO_4)^2 = (k_{nssSO4/MSA} \, \Delta MSA)^2 + (MSA\Delta \, k_{nssSO4/MSA})^2 + (\Delta SO_4)^2 \quad (6)$$

Uncertainties in the estimate of R values of sea-salt aerosol were then calculated as follows:

$$\Delta R^2 = (\Delta ssSO_4/Na)^2 + (ssSO_4 \times \Delta Na/Na^2)^2 \quad (7)$$

Results of the calculated R values of each impactor run are reported in Fig. 10 together with an estimate of the uncertainties. On a total of fifty runs, it was possible to evaluate R values on forty ones leading to a mean R value of $0.17 \pm 0.05$. As seen in Fig. 10, missed R values were due to either too low sodium, as in June/July 2007, or/and too high MSA, as in October

2009. Discarding two R values for which the error exceed 0.05 ($0.22 \pm 0.06$ in October 2009 and $0.27 \pm 0.13$ in September 2010), the average R value becomes $0.16 \pm 0.05$, suggesting that on average the sulfate depletion relative to sodium of sea-salt aerosol reaching Concordia in winter corresponds to a similar sea-salt fraction emitted from the open ocean (R=0.25) and from the sea-ice related processes (R ≈ 0.07).

The temporal variability of sea-salt aerosol reaching Dome C, its level and composition with respect to the sulfate relative to

sodium fractionation, were examined in the light of air masses reaching the site, as shown by calculated 10-day backward trajectories. The model was run every 6 h in backward mode for three different altitudes (0, 250, and 500 m above ground level, agl) for time periods corresponding to impactor run sampling. It is seen that when air masses arriving at Dome C have spent more than two days (up to 4 days) over oceanic sectors their mean sodium concentrations reach $47 \pm 21$ ng m$^{-3}$ against $12 \pm 5$ ng m$^{-3}$ when air masses have spent less than 0.5 days over oceans. Backward air mass trajectory calculations also

documented the time that air masses reaching Dome C were traveling within the marine boundary layer (below 600 m elevation) over oceans, distinguishing between sea-ice and open ocean areas. Selecting the impactor runs corresponding to sampling time periods over which the 10-day backward trajectories indicate more than one day of travel over the ocean, we find an overall decreasing trend of R (i.e. a stronger sulfate depletion relative to sodium) when air masses arriving at Dome C (0 m agl) have a longer contact with sea-ice than with open-ocean ($R^2 = 0.4$). Comparison of trajectories arriving at each





site at 0, 250 m, and 500 m agl, revealed similar data. Some typical examples of 10-day backward trajectories corresponding to strong, mid and weak sulfate relative to sodium fractionations are reported in Fig. 11. For example, from 9 to 23 September 2006, air masses reaching Concordia had travelled more than two days over oceanic sectors, with 40% and 60% of travel time within the boundary layer (below 600 m elevation) over the ice and open-ocean boundary, respectively. The

corresponding R value is low (R = 0.1, Fig. 11a), consistently suggesting a very significant contribution occurrence of sea-ice-related processes to the sodium level. At the opposite, from 28 August to 3 September 2008 (Fig. 11d), air masses arriving at Dome C have only travelled 5% of time within the boundary layer over the ice and (95% over open-ocean). Intermediate situations, from 26 July to 9 August 2007 (Fig. 11b) or 28 August to 11 September 2009 (Fig. 11c) are characterized by R values close to 0.15 and 20-25 % of travel time within the boundary layer over sea-ice (75-80% over

open-ocean).

A chemistry transport model, p-TOMCAT (parallelized-Tropospheric Offline Model of Chemistry and Transport), that includes open-ocean and blowing-snow sources was developed by Levine et al. (2014) to simulate sea-salt levels in Antarctica. Tested against atmospheric sea-salt observations, the model confirmed the importance of sea-ice-related sea-salt emissions in winter at both coastal and central Antarctica. Recently, a more accurate comparison of model simulations and

observations was done for the central site of Dome C (Legrand et al., 2016) on the basis of a 4-year record of bulk aerosol (2008-2011) at Concordia. We here discuss these 2008-2011 simulations with our observations. In summer, values close to 5 ng m$^{-3}$ of sodium were simulated, consistent with our 2006-2015 observations (Fig. 4b). In winter, simulations indicated higher sodium concentrations (~ 15 ng m$^{-3}$) from June to September. As seen in Fig. 4 b, whereas the 2006-2015 record also suggests a winter sodium maximum close to 15 ng m$^{-3}$, it has to be noticed that the maximum is observed slightly later (in

October and November). The model also suggests that a similar fraction of winter sea-salt aerosol at Concordia comes from open-ocean and sea ice. As above discussed, based on 38 impactor runs done in winter from 2006 to 2011, the observed R values (0.16 ± 0.05) suggests that, on average, the sea-ice and open-ocean contributions are similar at that season, consistent with the p-TOMCAT 2008-2011 simulations.

The mean mass size distribution of sodium is reported in Fig. 12 showing, that whatever the season, 80% of the sodium mass

is present on particles whose diameter ranges between 0.3 and 2.6 μm, the distribution peaking between 1 and 1.6 μm diameter. These observations are consistent with those from Udisti et al. (2012) showing a maximum between 1.1 and 2.1 μm diameter. However the finding of a larger size mode in summer (around 1-2 μm diameter) than in winter (submicrometric mode) is not here confirmed by our data.

Interestingly, as seen in Fig 13, in winter the sulfate depletion relative to sodium is less important in particles of which the

diameter is larger than 1.6 μm diameter. A slight increase of the sulfate depletion in particles smaller than higher than one micron diameter is also detected. Note that very rare are the cases for which such an examination of the sulfate depletion versus the size can be conclusive with respect to error uncertainties in the estimated R values; only sampling corresponding to a high sea-salt load permits that. If confirmed such a larger depletion of sulfate relative to sodium on the smaller than on the larger particles reaching Concordia in winter suggests that the process related to sea-ice involved in the production of





sea-salt aerosol in winter produces more small particles than the more common open-ocean bubble bursting process. This is also in the line with the observation made by Legrand et al. (2016) of a larger presence of sub-micron sea salt particles in winter than in summer at the coastal East Antarctic site of Dumont d'Urville, also broadly captured by model simulations, suggesting that the mechanism of sea-salt aerosol formation via sublimation of blowing salty snow particles, as formulated in

Yang et al. (2008), is reasonable.

### 3.3. Implications for ice core studies

This multiple-year round size-segregated aerosol studies conducted in central Antarctica provides a more statistically reliable estimate of the respective contribution of sea-salt emissions from the open ocean and sea-ice surface than previous studies based on a few impactor runs for which uncertainties induced by the difficulty to separate sea-salt and biogenic sulfate were

not accurately quantified. It is found that under present-day climatic conditions, a similar fraction of winter sea-salt aerosol at Concordia comes from open-ocean and sea-ice. Given the much larger extension of sea-ice, particularly during winter, around Antarctica during the last glacial maximum (Gersonde et al., 2005), it seems likely that the fraction of sea-salt aerosol from sea-ice was much increased at that time.

Both sea-salt aerosol and HCl present in the lower atmosphere of central Antarctica are expected to be efficiently trapped in

freshly deposited snow there. In the present-day (2006-2015) aerosol at Concordia, we observe an annual mean level of sodium of $6.9 \pm 11$ ng m$^{-3}$ and of chloride of $4.7 \pm 15$ ng m$^{-3}$, the corresponding grand average of r being close to 0.7 and the chloride depletion to 8.4 ng m$^{-3}$. The calculation of the chloride depletion was done by using equation (2) and by assuming a $k_{Cl/Na}$ value of 1.9 (i.e. 1.8 in summer and 2.0 in mid-winter, see Sect. 3.1). As discussed in section 3.1, if considering a sampling artefact leading to an overestimation of the annual mean lost of chloride by 40% due to acidification of HV filters,

the chloride depletion would be close to 6 ng m$^{-3}$ and the r value to 0.86. In the present-day atmosphere at Concordia, the mean annual level of HCl ($16 \pm 12$ ng m$^{-3}$, see Sect. 3.1) makes this gaseous species the dominant chloride species (three times more abundant than chloride present in the aerosol phase). The ratio of total chloride (aerosol plus HCl) to sodium of 3 in the present-day atmosphere is actually considerably lower than the value of ~8-10 in surface snowpack at Dome C (Röthlisberger et al., 2003), suggesting that initially HCl is more easily deposited than aerosol chloride. However, this

atmospheric chloride partitioning between gas and aerosol phase is subsequently modified after deposition as suggested by the observed decreasing trend of r values from 8-10 nearby the surface to less than one below 4-5 m below the surface (Legrand and Delmas, 1988; Röthlisberger et al., 2003). In ice deposited under present-day conditions (from 0 to 8 kyr BP) at Concordia, the mean sodium and chloride levels are of 22 and 15 ppb, respectively, and a mean r value of 0.67 is reported by Röthlisberger et al. (2003) for this time period. From this similarity of the r values in ice deposited between 0 and 8 kyr

BP and in present-day atmospheric sea-salt aerosol, and as previously mentioned in Sect. 3.1, we can conclude that most of HCl initially present in snow deposited at the site was lost to the atmosphere and chloride archived in snow is mainly related to sea-salt aerosol as it reaches the site.



In the ice deposited during the last glacial maximum (LGM, from 18 to 23 kyr BP), from the observation of a r value of 1.73, Röthlisberger et al. (2003) suggested that the de-chlorination of sea-salt aerosol was suppressed in the atmosphere at that time due to high dust levels that neutralized $HNO_3$ and $H_2SO_4$. Due to reduced snow accumulation rate during the colder climate of the LGM with respect to present-day conditions, we may expect that any HCl initially trapped in snow was

efficiently reemitted to the atmosphere. The examination of the r value in the LGM ice would therefore be related to the state of sea-salt aerosol reaching Concordia at that period. As discussed in Sect. 3.1, a $k_{Cl/Na}$ value of 1.8 and of 2.0 in summer and winter, respectively, adequately described the present-day freshly emitted sea-salt aerosol. If we assume a similar $k_{Cl/Na}$ value of 1.9 for glacial conditions than for present-day conditions, the observed 175 ppb of chloride and 100 ppb of sodium in ice deposited between 18 and 23 kyr BP (Röthlisberger et al., 2003) correspond to a lack of chloride relative to sodium of ~15

ppb. This value is likely underestimated since the expansion of sea-ice area in winter that reached a factor of two with respect to present-day (Gersonde et al., 2005) would lead to a higher value of $k_{Cl/Na}$ value of freshly emitted sea-salt in winter, possibly up to 2.2. If we assume $k_{Cl/Na}$ value of 2.0 for glacial conditions (i.e. 1.9 in summer and 2.1 in mid-winter), a lack of chloride relative to sodium of ~27 ppb is obtained (against 27 ppb in ice deposited between 0 and 8 kyr BP). Because atmospheric concentrations at Dome C are expected to be related to the deposition flux rather than the snow concentration,

the ~50% lower snow accumulation rate at the LGM suggests that the absolute depletion in the atmosphere was about a factor of 2 lower in the LGM than the Holocene. However there was a much reduced percentage loss (r value of 1.74 suggests 13% loss, compared to r value of 0.58 for the late Holocene representing 70% loss).

As seen in Figs. 4d and 4f, to-day, most of the nitrate is present in the Dome C atmosphere as nitric acid. The vast increase in nitrate concentration in Dome C ice in cold periods, extremely well-correlated with calcium (e.g., Legrand et al., 2000;

Wolff et al., 2010), at least suggests the possibility that nitrate may have been mainly attached to dust aerosol, with an absence of nitric acid over the polar plateau. This is difficult to assess, because much of the nitrate over the plateau today is secondary, created by photochemistry occurring in the snowpack (Savarino et al., 2007; Davis et al., 2008), and it is not known whether such photochemistry occurred or not during the LGM. A partial neutralisation of acidic sulfur aerosol by dust could also be envisaged.

However, this simple interpretation is hard to reconcile with measurements of the acidity of melted ice ($CO_2$ free measurements, Legrand et al., 1982), which revealed no significant difference in ice deposited during the LGM ($[H^+]$ = 1.9 ± 0.4 µEq $L^{-1}$) and the present-day climatic conditions ($[H^+]$ = 2.1 ± 0.5 µEq $L^{-1}$) (Legrand, 1987). This is not consistent with a simple change in acidity over Antarctica as the main reason for the relative absence of de-chlorination of sea-salt during the LGM, although it does not rule out a reduction of acidity over some part of the travel path or some parts of the year. In

particular, we can imagine that, if sea-ice was greatly extended, the proportion of sea-salt arriving in winter would be even more important and perhaps shifted earlier in the spring, at a time when sulfate aerosol and nitric acid concentration is much reduced.

It therefore seems possible that the estimated factor 2 reduction in the absolute amount of chloride depletion might be due to a reduced overlap between the period of high sea salt and the period of high nitrate and acidic sulfur aerosol, combined with




a possible partial neutralisation of nitric acid. However the much greater reduction in the percentage loss (which is what leads to r values close to that of sea-salt) requires another explanation, which is suggested by Fig. 7. Where grey bars extend above the sum of green and red, this can be taken to imply that chloride depletion is mainly limited not by the availability of sea-salt, but by the availability of nitric acid and acidic sulfur particles. During the LGM the sea-salt flux in Dome C ice

(taken as a measure of the sea-salt aerosol concentration) was increased by a factor ~2. The flux of $nssSO_4^{2-}$ remained almost constant (Wolff et al., 2010). The correlation of nitrate with calcium suggests that more nitrate was transported as aerosol, making it hard to assess the gaseous component alone, but Fig. 7 suggests that nitrate is anyway low during the months when sea-salt is high. This implies that the acidic aerosol and gas available for reaction with sea salt during the months when sea salt is highest was at best similar to today (and probably lower), while the amount of sea salt was much higher. This would

explain a modest reduction in the absolute value of Cl depletion and a much lower percentage depletion.

### 4. Concluding remarks

Load and composition of sea-salt aerosol at inland East Antarctica are established from multiple year-round records of bulk aerosol samplings, and for the first time in central Antarctica, the size-segregated composition of aerosol. These aerosol records are complemented, also for the first time over the high plateau, by multiple year-round sampling of gaseous HCl and

$HNO_3$ acids. It is shown that, except in mid-winter, a depletion of chloride relative to sodium takes place throughout the year, reaching a maximum of ~20 ng m$^{-3}$ in spring when, following winter, there are still large sea-salt amounts and acidic components start to recover. Acidic sulfur particles are always present in large enough amounts to replace chloride in small sea-salt particles whereas for larger particles gaseous nitric acid competes with acidic sulfur compounds. We find that HCl is around twice more abundant than the amount of chloride lost by sea-salt aerosol, suggesting that reemission from the snow

pack represents an additional significant HCl source over the Antarctic plateau. These atmospheric data on the chloride depletion with respect to sodium in aerosol and the amount of chloride present as HCl in the gas phase are shown to be useful in discussing the aging of sea-salt in the past as inferred from ice core chemical records. Impactor runs done in winter (from 2006 to 2011) indicate a mean sulfate to sodium ratio of sea-salt aerosol present over central Antarctica of $0.16 \pm 0.05$, suggesting similar contribution of sea-ice and open ocean emission to the sea-salt load over inland Antarctica. It is shown

that the sulfate depletion relative to sodium is larger when air masses arriving at Dome C had travelled a longer time over sea-ice than over open-ocean.

### Data availability

Data on aerosol (bulk and size-segregated composition) and gaseous nitric and hydrochloric acids at Concordia can be made available for scientific purposes upon request to the authors (contact: michel.legrand@univ-grenoble-alpes.fr or

Suzanne.Preunkert@univ-grenoble-alpes.fr).





**Acknowledgements**

National financial support and field logistic supplies for the summer campaign were provided by the Institut Polaire Français-Paul Emile Victor (IPEV) through program n°414 and 903 and the Agence Nationale de la Recherche through

contract ANR-14- CE01-0001-01 (ASUMA). This work was initiated in the framework of the French environmental observation service CESOA (Etude du cycle atmosphérique du Soufre en relation avec le climat aux moyennes et hautes latitudes Sud, http://www-lgge.obs.ujf-grenoble.fr/CESOA/spip.php?rubrique2) with the financial support of INSU (CNRS).

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




**Figures**

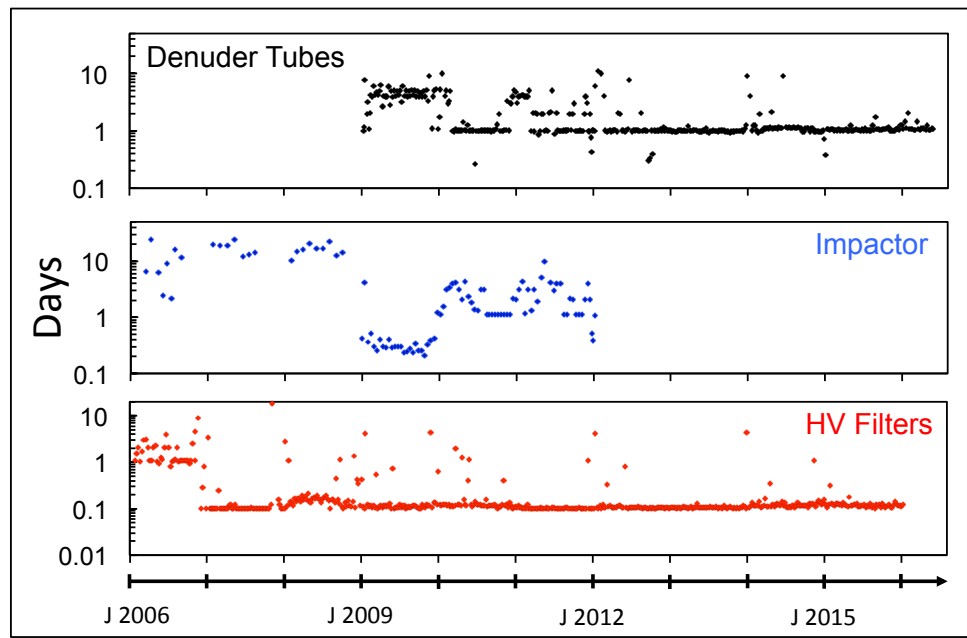

**Figure 1.** Time (in days, logarithmic scales) spent between two successive samplings for acidic gases (denuder tubes), size-segregated (12-stage impactor) and bulk (HV filters) aerosol composition at Concordia (2006-2015) (Sect. 2).





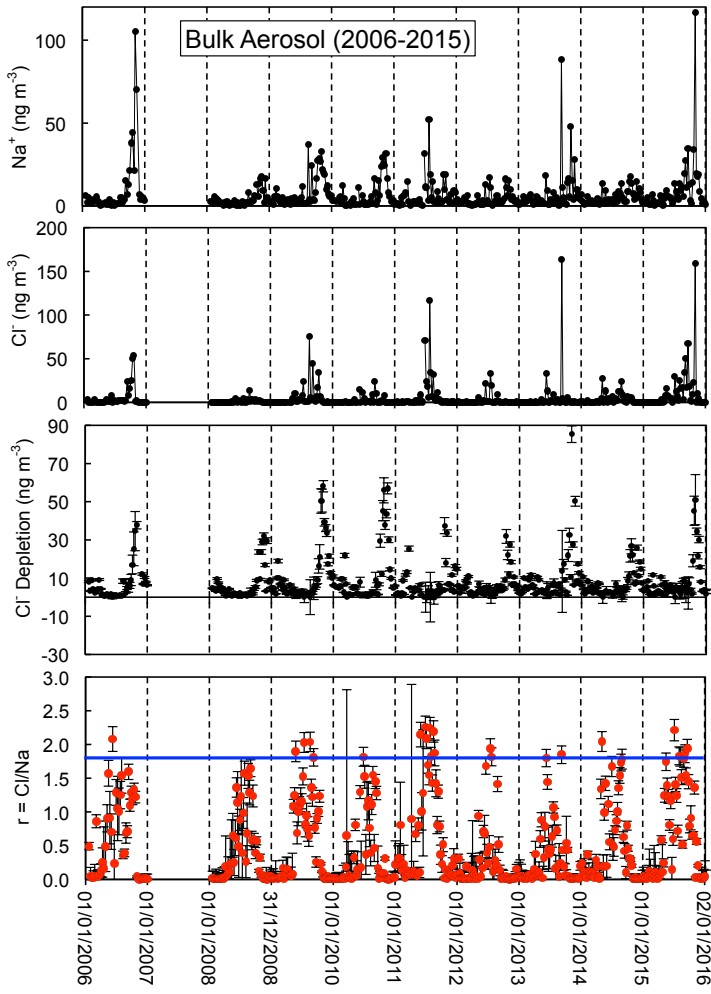

**Figure 2.** Weekly bulk aerosol concentrations of sodium and chloride and the calculated chloride depletion relative to sodium with respect to the composition of freshly emitted sea-salt aerosol (equation 2) and the mass Cl/Na ratio (r). Vertical bars refer to uncertainty in calculating r (equation 1) and the chloride depletion value (equations 3 and 4). The horizontal blue line refers to the r seawater reference value (1.8).



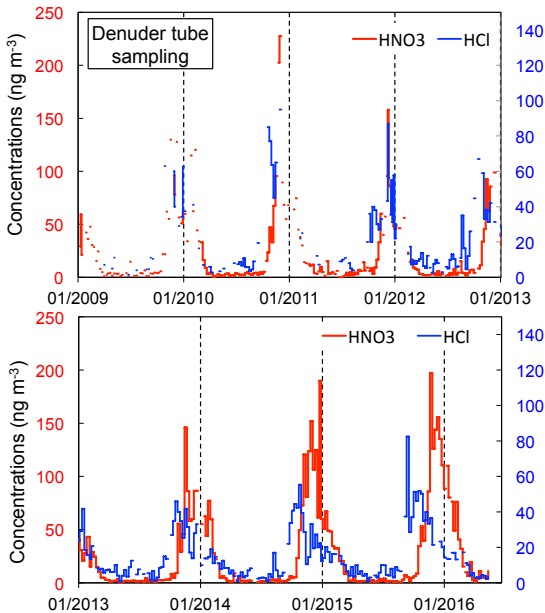

**Figure 3.** Year-round records of HCl and HNO₃ sampled on denuder tubes at Concordia from January 2013 to May 2016.



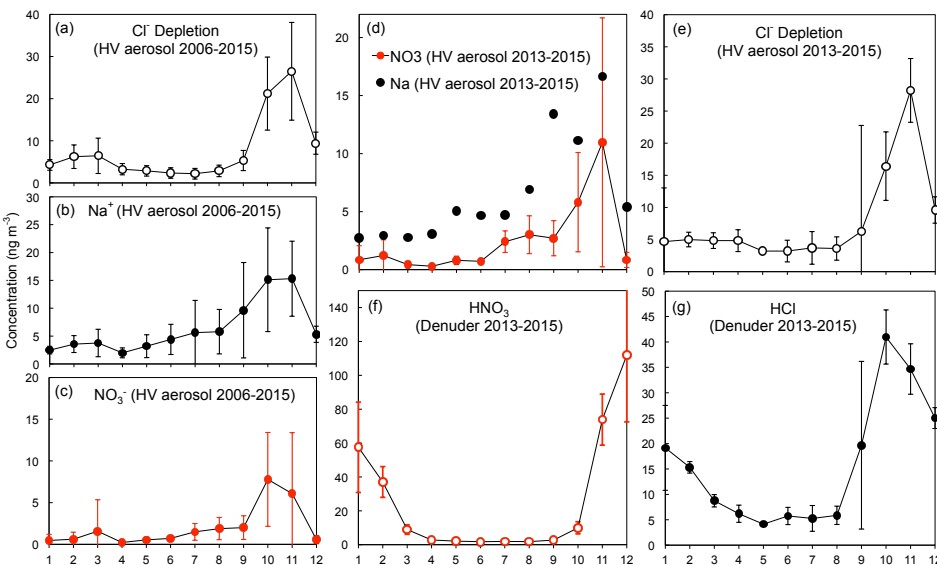

**Figure 4.** From (a) to (c): Monthly mean Cl⁻ depletion relative to Na⁺ with respect to the composition of freshly emitted sea-
10  salt aerosol (a), concentrations of Na⁺ (b) and NO₃⁻ (c) in bulk aerosol. From (d) to (g): Monthly mean Na⁺ and NO₃⁻
concentrations (d) and Cl⁻ depletion relative to Na⁺ with respect to the composition of freshly emitted sea-salt aerosol (e) in
bulk aerosol (2013-2015) together with concentrations of HCl (g) and HNO₃ (f) sampled on denuder tubes over the same
time period. Vertical bars refer to the year-to-year variability.





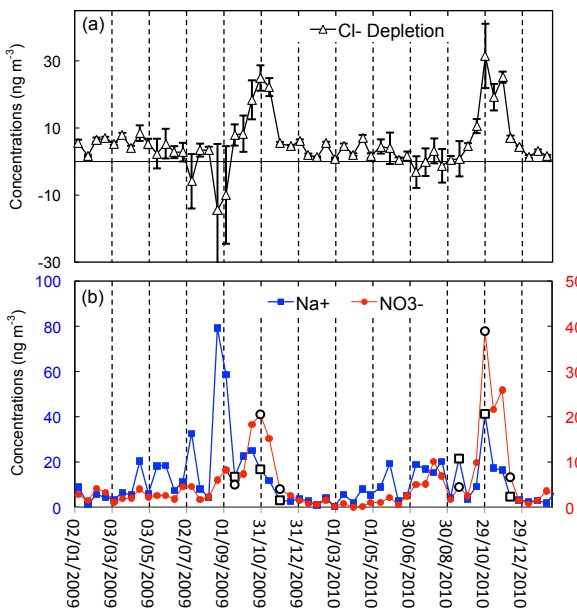

10 **Figure 5.** (a) Cl⁻ depletion relative to Na⁺ with respect to the composition of freshly emitted sea-salt aerosol (equation 2) and

(b) concentrations of $Na^+$ and $NO_3^-$ in aerosol collected on the 12-stage impactor in 2009 and 2010 (biweekly sampling).

Vertical bars refer to uncertainty in calculating the chloride depletion (equations 3 and 4).





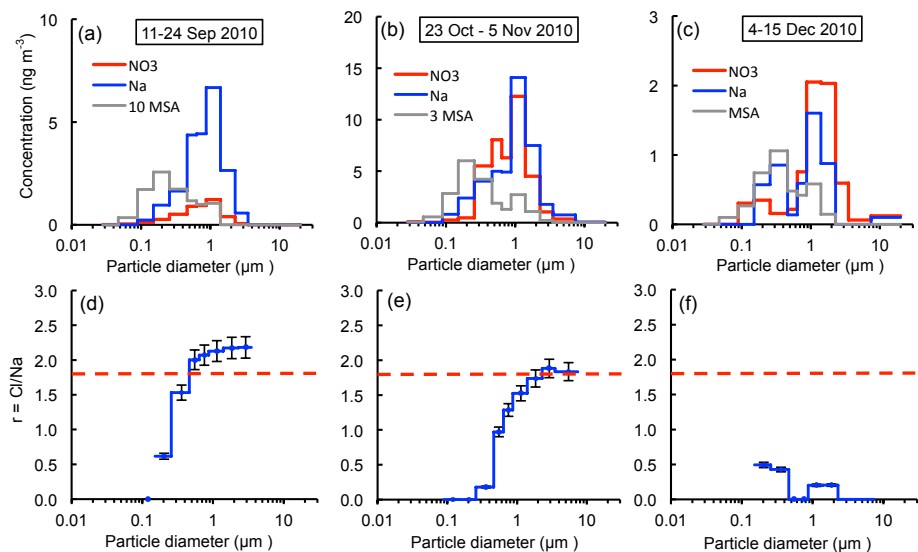

**Figure 6.** From (a) to (c): Size-segregated aerosol composition (nitrate, sodium, and MSA) from September to December 2010 at Concordia. In (a) and (b), MSA levels were multiplied by a factor of 10 and 3, respectively. From (a) to (c), the levels of $HNO_3$ simultaneously sampled on denuder tubes were enhanced from $3.5 \pm 1$ ng m$^{-3}$ to $40 \pm 7$ ng m$^{-3}$, and $80 \pm 1$ ng

10    m$^{-3}$ (see Sect. 3.1). From (d) to (f), the corresponding r values as function of the aerosol size.



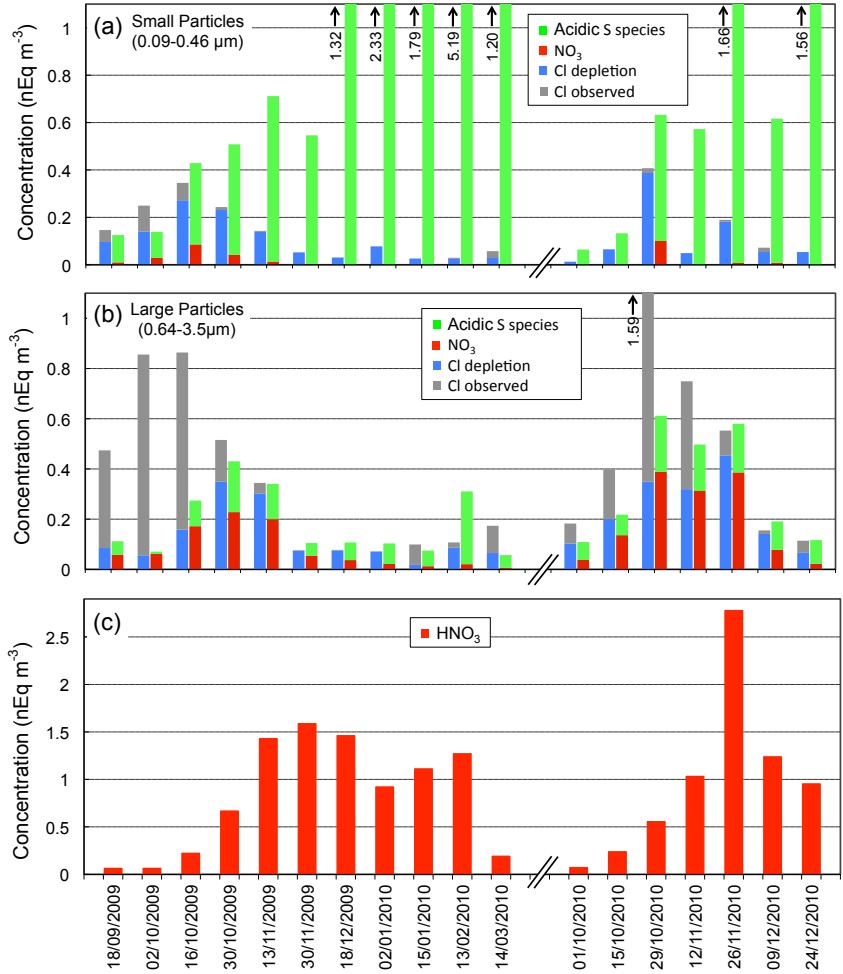

**Figure 7.** Acidic compounds and chloride depletion relative to sodium from October to December in 2009 and 2010. (a) and

(b): Observed chloride (grey) and estimated loss of chloride (in blue, equation 2) versus nitrate (in red) and acidic sulfur

5 species (in green, see Sect. 3.1) in small (a) and large particles (b). (c): $HNO_3$ sampled on denuder tubes over the same time

period.




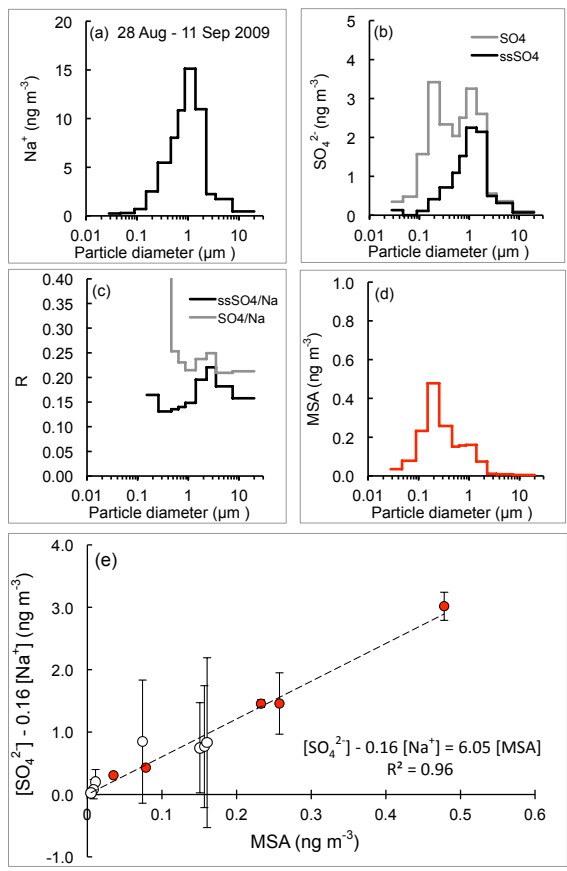

**Figure 8.** From (a) to (d) : Mass size distribution of $Na^+$ (a), $SO_4^{2-}$ (b, grey curve), and MSA (d) of the aerosol sampled at Concordia 28 August to 11 September 2009. Also plotted is the sulfate to sodium mass ratio (R) (c, grey curve). The black lines on the panels of sulfate and sulfate to sodium ratio (R) refer to values of sea-salt sulfate calculated after having subtracting the biogenic contribution estimated from MSA (see panel e). (e): Relationship between biogenic sulfate and MSA observed on the 12 stages of the impactor, the red points referring to values on the smaller particles (below 0.5 μm diameter). The vertical bars denote uncertainties in calculating the biogenic contribution by assuming a sulfate to sodium ratio in sea-salt aerosol ranging between 0.07 and 0.25 (0.16 ± 0.09) (Sect. 3.2).

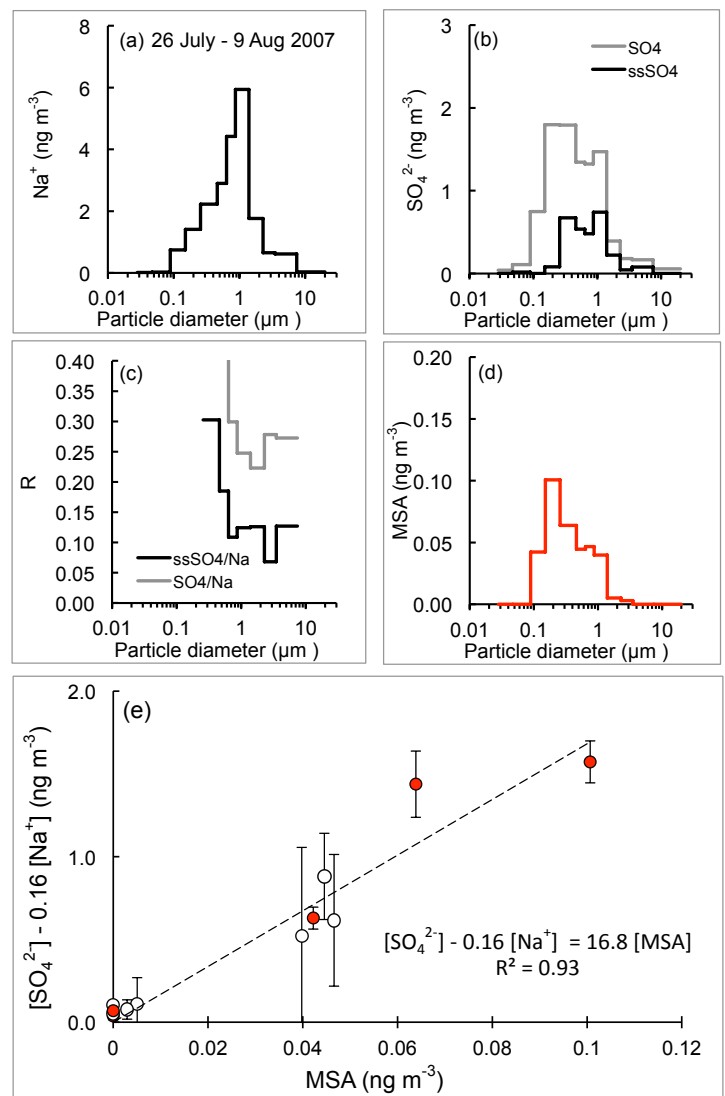

**Figure 9.** Same as Fig. 8 for aerosol collected between 26 July and 9 August 2007.





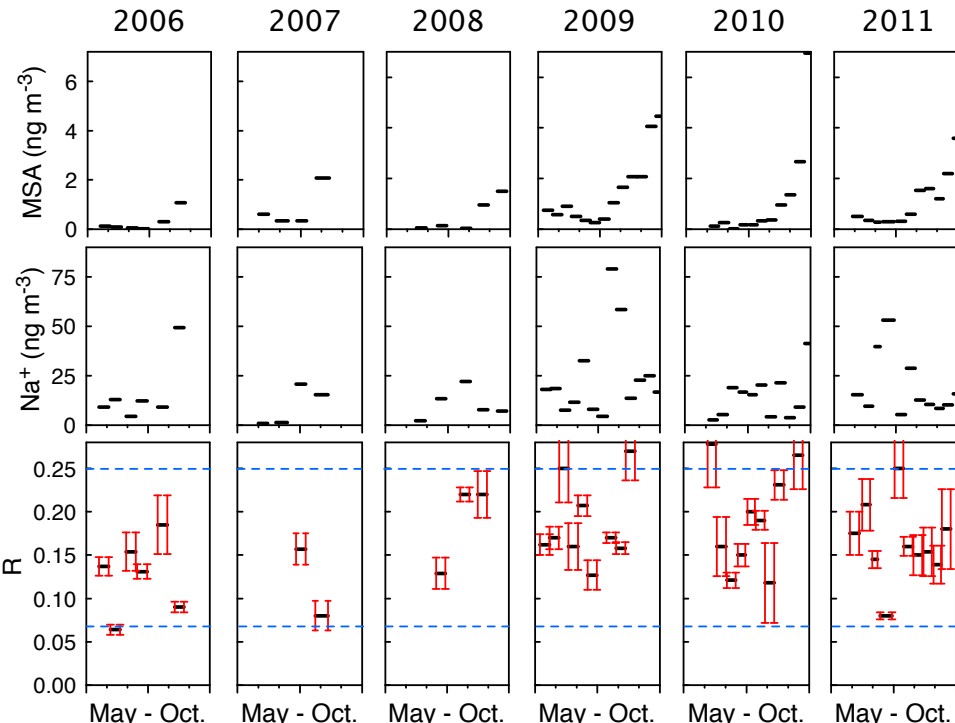

**Figure 10.** Year-round winter composition of aerosol collected on the 12-stage impactor sampler. MSA and sodium refer to the total concentration (i.e. the sum of concentrations observed on the 12 stages). R is the sulfate to sodium mass ratio of sea-salt aerosol. R values were derived from sulfate and sodium present on the impactor stages having collected most of sea-salt (from 0.5 to 2 µm diameter, see examples reported in Fig. 8 and 9) and for which sulfate concentrations have been corrected

10 from its biogenic fraction (equation 5). Vertical bars refer to uncertainties related to the estimation of R (equation 7).





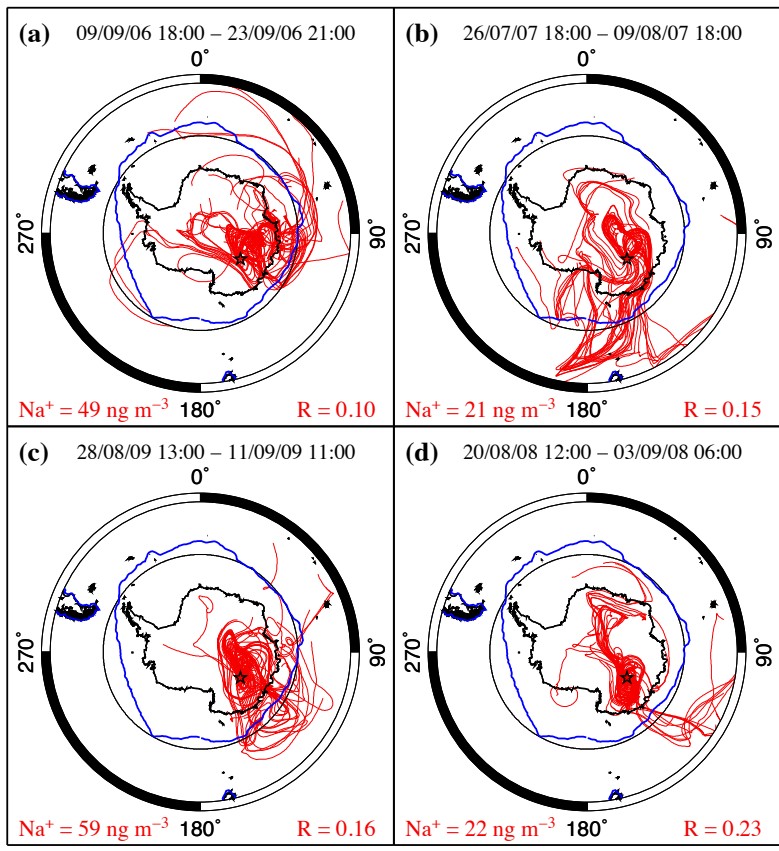

**Fig. 11.** Examples of 10-day backward trajectories corresponding periods of aerosol sampled by using the 12 stage impactor

5 and for which a strong to moderate and weak sulfate depletion relative to sodium was observed (see Fig. 10). All lines

correspond to arrivals at 0 m agl. The blue line refers to the mean location of the sea-ice edge end of winter (August) over

the period 1981–2012 NOAA_OI_SST_V2 data provided by the NOAA/OAR/ESRL PSD, Boulder, Colorado, USA, http://

www.esrl.noaa.gov/psd). In each case, we report in red the sodium concentration and the sulfate to sodium mass ratio (R)

related to sea-salt particles (see Fig. 10).





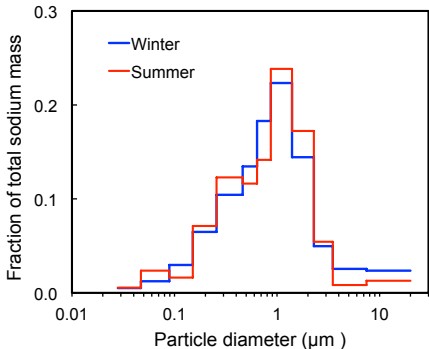

**Fig. 12.** Mean size-segregated mass of sodium (expressed as fraction of the total sodium mass) in winter (May-October, 55 samples) and summer (November-April, 40 samples).





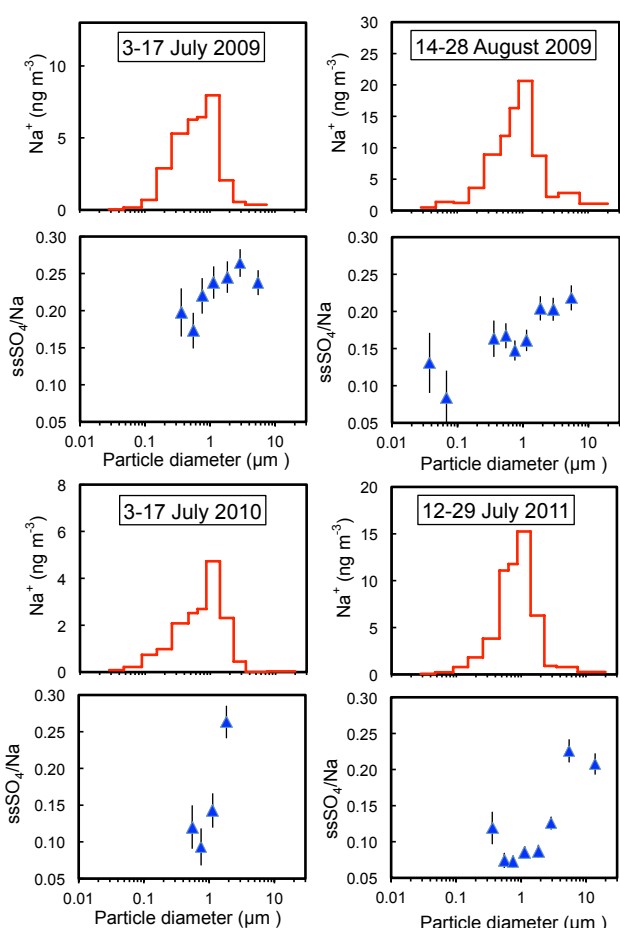

**Fig. 13.** Size distribution of sodium and sea-salt sulfate to sodium (R) ratio in 4 winter aerosol samples collected at Concordia. The vertical bars refer to uncertainties in calculating R.