# Peer review of "Year-round records of bulk and size-segregated aerosol composition in central Antarctica (Concordia site) Part 1: Fractionation of seasalt particles"

_Atmospheric Chemistry and Physics, 2017_

## Referee Comment (RC1) · Anonymous Referee #2 · 21 Aug 2017

Comments on Legrand et al. 'Year-round records of bulk and size-segregated aerosol composition in central Antarctica (Concordia site) Part 1: Fractionation of sea-salt particles'

**General comments:**

This manuscript presents a detail examination of chemical compounds, mainly depletion of sulfate and chlorine relative to sodium with respect to freshly emitted sea salt aerosol, in aerosols that arrived at Concordia in central Antarctica, as well as acidic gases (HCl and $HNO_3$) in air based on their multiple year-round records of samplings. Both bulk and size-segregated depletion of chloride and sulfate are reported with a major goal of determining the origin of sea salt aerosols reaching the high plateau. To achieve that, the authors have introduced a novel method to remove biogenic sulfate contamination to the sulphates containing in the aerosols. Their approach has been successfully applied to the raw dataset and the conclusion derived indicating rough similar contribution of sea salt aerosols from both sea-ice and open ocean emission during polar winter is well consistent with previous model findings. Together with a back-trajectory result, they further confirm that a larger sulfate to sodium ratio would be observed when air masses had travelled a longer time over sea ice than over open water. Moreover, their novel data seem to show that small sub-micron sized SSA has much larger sulfate depletion that larger ones which is likely to indicate different sized SSA could face quite different preferential production mechanisms as predicted previously.

The second major part of the manuscript is about chlorine depletion in aerosols. Their multi-year data clearly show a maximum depletion occurring in spring which is associating with high level of sea salt amount. The interpretation of the spring peak in chloride depletion has been discussed carefully, in associating with concentrations of nitrate, acidic sulphur in aerosols, as well as nitric acid and HCl in gas phase.

In general, this is a novel and deepening study of aerosol depletions for chlorine and sulphate at a central Antarctica site. It is well written and with huge scientific implications for our understanding of the sea salt ice records taken from inland Antarctica. It deserves a publication in ACP with a minor revision (see my specific comments shown below).

**Specific comments:**

P12: In terms of the chlorine depletion mechanisms, only two controlling factors (e.g. nitrate and sulphate) are highlighted in the manuscript. Given the fact that the largest chlorine depletion occurs in spring and is associating with highest SSA load, a season seeing the highest BrO in coastal sites of the Antarctic, then the manuscript would benefit if the authors could discuss the potential role of halogen, here mainly bromine in influencing cholrine release from SSA. Particularly, chloride can be liberated via heterogeneous reaction through $HOBr(g)+Cl^- \rightarrow BrCl$ (e.g. see Abbatt et al., 2012). Though at inland sites like Concordia, this process is not important due to the relatively low inorganic bromine species (a few pptv, see their paper of Legrand et al., 2016), however, at coastal sites during bromine explosion events in spring, the boundary layer HOBr concentration could be as high as (or even higher than) $HNO_3$. Apart from this process, $BrONO_2$ hydrolysis reaction (via heterogeneous

reaction on wet surface) has been found dominating nitric acid formation in high latitude. For instance, a global model (pTOMCAT) with a detailed bromine chemistry included clearly showed that when BrONO2 hydrolysis is included, about 60-80% atmospheric NOx in high latitudes will be removed (Fig. 12c of Yang et al., 2005) indicating $BrONO_2$ hydrolysis could efficiently convert NOx to $HNO_3$ in mid- to high latitude (not in low latitude due to high OH and low BrO there). Does this process play a role, either directly or indirectly, in affecting chlorine depletion? Can bromine be ruled out completely as a role in $HNO_3$ formation and chlorine depletion? If not, then a discussion should be given.

P24, Figures 6/7: My second concern is again about the chlorine depletion in section 3.1. If the acidic replacement is indeed dominating chlorine release, them it could be possible to see a nearly 1:1 linear scatter relationship between chlorine depleted and $[H^+]$ derived. The $[H^+]$ can be derived from the two major acidic species (nitrate and sulphate without SS_SO4). Equally, a scatter plot between the ratio of r and ratio of $[H^+]/[Na^+]$ is also indicative to whether the replacement process is working. This examination can be made both for seasonal depletion data shown in figure 7 as well as for size-dependent depletion shown in figure 6. Instead of using $[H^+]$, a sum of $[NO_3]/M_{NO3}+2X[SO_4]/M_{SO4}$ can also be used in the scatter plot. here $M_{NO3}$ and $M_{SO4}$ are molecular weight of $NO_3$ and $SO_4$ respectively.

**Technical corrections:**

P2L32: add 'mass' between 'sodium' and 'ratios'

P8L10: remove 'to' before 'due to'

P8L20: 'Fig.6' should be 'Fig. 7' ?

P8L18: 'larger' here should be 'smaller'?

P9L19: remove the extra 'and' in 'or/and and too high'

P12L3. A reference is needed to 'high dust levels that neutralized $HNO_3$ and $H_2SO_4$' .

P12L29: change 'part' to 'parts'

In most of figures, change NO3 to $NO_3^-$, SO4 to $SO_4^=$ , Na to $Na^+$.

P22 figure 4: it would be good to add 'r' line to high light the 'relative depletion factor.

P24 figure 6. Please add a line of total sulphate derived based on MAS (without SS_SO4).

P25 figure 7b: is the number '0.64' in '(0.64-3.5 µm)' actually 0.46? why there is a gap between 0.46 to 0.64 µm ? same question in P6 L25.

P28: what do the two dash lines represent for?

**References:**

Abbatt, J. P. D., J. L. Thomas, K. Abrahamsson, C. Boxe, A. Granfors, A. E. Jones, M. D. King, A. Saiz-Lopez, P. B. Shepson, J. Sodeau, D. W. Toohey, C. Toubin, R. von Glasow, S. N. Wren, and X.

Yang, Halogen activation via interactions with environmental ice and snow, Atmos. Chem. Phys., 12, 6237-6271, [doi:10.5194/acp-12-6237-2012] 2012.

Legrand, M., X. Yang, S. Preunkert, and N. Theys (2016), Year-round records of sea salt, gaseous, and particulate inorganic bromine in the atmospheric boundary layer at coastal (Dumont d'Urville) and central (Concordia) East Antarctic sites, J. Geophys. Res. Atmos., 121, 997–1023, doi:10.1002/2015JD024066.

Yang, X.; Cox, RA; Warwick, N. J.; Pyle, J. A.; Carver, Glenn D.; O'Connor, FM; Savage, NH; Tropospheric bromine chemistry and its impacts on ozone: A model study. *Journal Of Geophysical Research*, 110 (D23311), doi:10.1029/2005jd006244, 2005.

---

## Referee Comment (RC2) · Anonymous Referee #3 · 12 Sep 2017

The manuscript presents multi-year measurement results of seal salt aerosols and associated ion composition as well as HCl and HNO3 acid gases in the central Antarctic site. With these data, the authors examine the chloride depletion relative to sodium with respect to freshly emitted sea salt aerosols and the sulfate depletion relative to sodium with respect to the composition of sea water. The seasonal variability of such depletion, the role of acidic sulfur aerosol and nitric acid in the depletion, and the contribution sea-ice and open ocean emissions to the sea salt aerosols load are investigated. The reported data are valuable and such kind of study should be a welcome addition

to the literature on Antarctic environmental and atmospheric chemistry research. On the other hand, the manuscript appears to be not well written and some discussions are ambiguous without a clear clue to follow. Especially there is a lackness of indepth analysis based on the fundamental chemistry. In my opinion the quality of this manuscript is not high enough to be published in ACP for its current version. Below are my comments in detail.

What does the authors mean in the term of "acidic sulfur aerosol" (e.g., in P1, L22)? The authors also define nitrate as "acidic compounds" (P6, L26-28). Do they refer to HNO3 in aerosols? Note that SO4= and NO3- should be taken as neutral, instead of acidic, ions in water solution. They calculated the acidic sulfur component as the sum of non-sea-salt sulfate plus MSA after subtracting the amount of ammonium (P6, L27-28). What does the residual mean in acid-base equilibrium chemistry? The authors should have investigated the balance status between cations and anions in measured aerosols before analyzing the chlorine depletion by so called replacement reactions. Figure 7 and related discussions (P7, L10-17) provide some information. However, only when the full set of ions have been taken into account and, if possible, neural compounds (e.g., CaSO4 , Na2SO4, (NH4)2SO4, and NaNO3) diagnosted, the displacement process could be understood clearly. Note that there is a neutralization order for the ions in a solution, e.g., SO4= > NO3- for anions and Ca2+ > Na+ for cations. From Figure 7c, one cannot see whether the displacement had happened or not. Only by investigating NO3-, Na+ and other associated ions in the aerosols, the role of HNO3 in the displacement might be seen. Note that partitioning of HNO3 between gaseous and aerosol phases depends not only on the acidity of liquid aerosols but also atmospheric temperature.

For Section 3.2 of the manuscript, neither the terms of SO4, biogenic sulfate, ssSO4, nssSO4 and MSA and are well defined, nor their measuring (or calculating ) methods clearly introduced. It is difficult for me to follow the discussions as the assumption (e.g., the relationship between biogenic sulfate and MSA in P8, L22-25) has not been

fully based one fundamental atmospheric chemistry and physics. There is a doubt that the regression results from small particles can be applied to large ones. The authors refers to a companion paper: "Legrand, M., Preunkert, S., Weller, R., Zipf, L., Elsässer, C., Merchel, S., Rugel, G., and Wagenbach, D: Year-round (2006-2015) record of bulk and size-segregated aerosol composition in central Antarctica (Concordia site) Part 1: Sulfur derived aerosol (MSA and sulfate), this issue" (note that both companion papers are labelled as "Part 1"). Considering that sulfate aerosol is a large and important part of this manuscript, I would suggest the paper not to be separated into two parts. Sulfate depletion relative to sodium has been attributed to sea-ice related emissions due to precipitation of mirabilite ($Na_2SO_4.10H_2O$) during freezing of seawater. As both sulfate and sodium will loss with the deposition of mirabilite, how the depletion of sulfate relative to sodium occurs should be explained quantitatively in detail.

The manuscript needs to be concise and formulas (1), (3), (4), (6), and (7) and associated descriptions can be moved to the Supplement. The authors use the p-TOMCAT model to confirm their conclusion on the source of sea salt aerosols (P10, L11-23). But the model and its simulation results have not been well evaluated and introduced in this manuscript. These discussions (P10, L11-23) provide no more convincing information than the backward trajectory analysis described in previous paragraph. I do not think that Sect 3.3 should be included in this manuscript as it appears to provide no strong support to explain the observational results of this study.

---

## Referee Comment (RC3) · Anonymous Referee #4 · 13 Sep 2017

The authors address a pressing topic; sea salt, its size distribution in aerosols and co-located gas phase measurements over annual cycles. The interpretation of sea salt records in Antarctica is relevant to sea ice distribution patterns recorded in glacial records. This manuscript contributes significantly to this discussion, demonstrating that approximately half the sea salt aerosol load comes from open ocean versus sea ice. The relationship of sea salt, aerosol acidification and the production of HCl and HNO3 is also examined by considering denuder measurements of acid gases and their effect on aerosol characteristics. The authors argued that HCl emission from surface snow

or differences in the lifetime of HCl versus sea salt transported from the coast drove differences observed between denuder and aerosol measurements. These results are novel and intriguing. This paper provides constraints to some long-standing questions regarding the source of anomalies in the ratio of Na to Cl in snow, whether and by how much Cl in aerosols is depleted in the continental Antarctic and in ice and the factors driving it. These points are all strong and convincing. Arguments about sea salt and biogenic sulfate however, are less well-rounded: likely this is because detail is provided in a separate manuscript dealing with sulfate and sulfate to sodium ratios. The manuscript is strong, but some sections, particularly with respect to sulfate and sulfate to sodium ratios need additional attention prior to publication.

Portions of this manuscript associated with interpretation of sulfate needs additional description (that perhaps can be found in the accompanying paper on sulfur-derived aerosol) to be fully understandable. Page 8 line 24 refers to quantifying biogenic sulfate using MSA and the relationship between MSA and sulfate on particles with a diameter <0.46 microns. This assumes that all non sea salt sulfate in aerosols <0.46 microns in diameter is biogenic. Is this because anthropogenic sulfate is negligible and volcanic sulfate is absent in the Antarctic interior? A previous study in the Arctic demonstrated that MSA in for aerosols <0.49 microns correlated with the presence of fine particles rather than with biogenic sulfate (Rempillo et al., 2010). In that study the surface of <0.49 micron aerosol, rather than biogenic sulfate, were the dominant factor affecting MSA formation. The authors need to demonstrate why anthropogenic and volcanic sulfate are not factors in MSA formation at this continental Antarctic site. Line 23 on page 8 ".. we therefore have corrected the concentrations of sulfate present on the upper stages of the impactor for the (nss rather than biogenic contribution)".

On page 10 how were the values of 40% and 60% of travel time within the boundary layer determined? This is not evident from the examples of air mass back trajectories provided in Figure 11. Is there a reference or data to support a boundary layer of ∼600 m for open ocean and over sea-ice in the Antarctic at this time of year? The discussion

suggests a quantified analysis that is not shared with the reader which makes this section less convincing.

Line 30 on page 10 is confusing "A slight increase of the sulfate depletion in particles smaller than higher than one micron. . .."

---

## Editor Comment (EC1) · J. Ma (Editor) · 13 Sep 2017

Dear Dr. Legrand,

My apologize for such a long time for the open discussions of your manuscript. Your manuscript did undergo an unusual experience, which is certainly unpleasant for you as well as for us. Two referees had accepted our invitation to review your manuscript in April. But both of them failed to submit their reports in early July when the open discussions should be closed normally for your manuscript. It took a few weeks more

for me to contact them and to nominate other potential referees. Although I found another two referees agreeing to review your manuscript, unfortunately the report from one referee was missing again by the initial and extended deadlines. While promising to submit the review report soon, one referee said in his/her personal email to me that the paper is good but tough to get through and it's taken longer than expected. Actually, I share the same feeling with that referee when reading your manuscript.

Now we have gotten two review reports. While both referees admire import value of your data and significance of your work, one of them rates a low value of the quality of your manuscript especially in presentation. I agree with the referee (Referee 3) in that the manuscript needs to be focused more on the analysis of chemical processes. Actually, another referee (Referee 1) also suggested that additional chemical process be considered for chlorine depletion relative to sodium with respect to freshly emitted sea salt aerosols.

I noted that the sulfate aerosol issue has been intensively addressed by a companion paper of this manuscript (Legrand et al., 2017), which was also published in ACPD. Therefore, you may refer to that paper for the filtering of biogenic sulfate aerosols and, as suggested by the referee, focus more on the ionic chemistry involved in sulfate depletion relative to sodium with respect to the composition of sea water.

I also agree with the referee in that the discussions on implications for ice core studies (Sect. 3.3) should be skipped over if these discussion help little to explain your measurement and analysis results presented in the preceding sections.

In summary, I think that your manuscript needs substantial revisions based on the comments from the referees. You are welcome to submit the revised manuscript if you think that all the issues they raised can be well addressed. Your manuscript will be sent to the referees for further review, and the final decision can be made then.

If you have any questions, please do not hesitate to contact me.

Sincerely,

Jianzhong Ma

Reference: Legrand, M., Preunkert, S., Weller, R., Zipf, L., Elsässer, C., Merchel, S., Rugel, G., and Wagenbach, D.: Year-round record of bulk and size-segregated aerosol composition in central Antarctica (Concordia site) Part 2: Biogenic sulfur (sulfate and methanesulfonate) aerosol, Atmos. Chem. Phys. Discuss., 2017, 1-39, 10.5194/acp-2017-305, 2017.

---

## Author Comment (AC1) · 16 Sep 2017

Response to Reviewer 1:

**General comments:**
This manuscript presents a detail examination of chemical compounds, mainly depletion of sulfate and chlorine relative to sodium with respect to freshly emitted sea salt aerosol, in aerosols that arrived at Concordia in central Antarctica, as well as acidic gases (HCl and $HNO_3$) in air based on their multiple year-round records of samplings. Both bulk and size- segregated depletion of chloride and sulfate are reported with a major goal of determining the origin of sea salt aerosols reaching the high plateau. To achieve that, the authors have introduced a novel method to remove biogenic sulfate contamination to the sulphates containing in the aerosols. Their approach has been successfully applied to the raw dataset and the conclusion derived indicating rough similar contribution of sea salt aerosols from both sea-ice and open ocean emission during polar winter is well consistent with previous model findings. Together with a back-trajectory result, they further confirm that a larger sulfate to sodium ratio would be observed when air masses had travelled a longer time over sea ice than over open water. Moreover, their novel data seem to show that small sub-micron sized SSA has much larger sulfate depletion that larger ones which is likely to indicate different sized SSA could face quite different preferential production mechanisms as predicted previously.

The second major part of the manuscript is about chlorine depletion in aerosols. Their multi-year data clearly show a maximum depletion occurring in spring which is associating with high level of sea salt amount. The interpretation of the spring peak in chloride depletion has been discussed carefully, in associating with concentrations of nitrate, acidic sulphur in aerosols, as well as nitric acid and HCl in gas phase.

In general, this is a novel and deepening study of aerosol depletions for chlorine and sulphate at a central Antarctica site. It is well written and with huge scientific implications for our understanding of the sea salt ice records taken from inland Antarctica. It deserves a publication in ACP with a minor revision (see my specific comments shown below).

**Specific comments:**

P12: In terms of the chlorine depletion mechanisms, only two controlling factors (e.g. nitrate and sulphate) are highlighted in the manuscript. Given the fact that the largest chlorine depletion occurs in spring and is associating with highest SSA load, a season seeing the highest BrO in coastal sites of the Antarctic, then the manuscript would benefit if the authors could discuss the potential role of halogen, here mainly bromine in influencing cholrine release from SSA. Particularly, chloride can be liberated via heterogeneous reaction through $HOBr(g)+Cl^- \rightarrow BrCl$ (e.g. see Abbatt et al., 2012). Though at inland sites like Concordia, this process is not important due to the relatively low inorganic bromine species (a few pptv, see their paper of Legrand et al., 2016), however, at coastal sites during bromine explosion events in spring, the boundary layer HOBr concentration could be as high as (or even higher than) $HNO_3$. Apart from this process, $BrONO_2$ hydrolysis reaction (via heterogeneous reaction on wet surface) has been found dominating nitric acid formation in high latitude. For instance, a global model (pTOMCAT) with a detailed bromine chemistry included clearly showed that when BrONO2 hydrolysis is included, about 60-80% atmospheric NOx in high latitudes will be removed (Fig. 12c of Yang et al., 2005) indicating $BrONO_2$ hydrolysis could efficiently convert NOx to $HNO_3$ in mid- to high latitude (not in low latitude due to high OH and low BrO there). Does this process play a role, either directly or indirectly, in affecting chlorine depletion? Can bromine be ruled out completely as a role in $HNO_3$ formation and chlorine depletion? If not, then a discussion should be given.

*Concerning the direct role of HOBr on the release of chloride from sea-salt, its level even in spring at the coast is limited to 1 pptv or so (Legrand et al., 2016). For comparison the $HNO_3$ level at Concordia reaches 0.5 nEq $m^{-3}$ (i.e., ~11 pptv) and more than 1 nEq $m^{-3}$ in November-December (i.e., more than 22 pptv). Concerning coastal regions, in samplings made on denuder tubes at DDU, Legrand et al. (2016) reported mass ratios of $Br/NO_3$ of 0.2 in summer and close to 1-2*

*in winter. Checking these data we calculate a mean ratio of 0.7 for spring (September/October). That leads to a molar ratio of 0.5. Since HOBr represents at the east coast a quarter of the total inorganic bromine (Legrand et al., 2016, Figure 14), we can conclude that, at least in East Antarctica, the role of HOBr cannot be ruled out but is certainly not dominant. We have introduced this discussion in the text:*

*"Several previous studies discussed the nature of chemical species (nitric acid, sulfuric and methanesulfonic acid) involved in the dechlorination of sea-salt aerosol in Antarctica but no overall picture yet emerged. Chloride can also be released from the reaction of gaseous HOBr with sea-salt aerosol (Abbatt et al., 2012). The HOBr level at Concordia was investigated by Legrand et al. (2016) who reported mixing ratios close to 1 pptv in spring. For comparison, the $HNO_3$ level at Concordia reaches 0.5 nEq $m^{-3}$ (i.e., ~11 pptv) in October and more than 1 nEq $m^{-3}$ in November-December (i.e., more than 22 pptv), suggesting that HOBr does not significantly contribute to the chloride depletion over the high Antarctic plateau in spring and summer. At coastal regions, in samplings made on denuder tubes at DDU, Legrand et al. (2016) reported a bromide to nitrate mass ratio of 0.2 in summer and close to 1-2 in winter (a mean ratio of 0.7 being observed for September/October). That leads to a bromide to nitrate molar ratio of 0.5. Since, at that site, HOBr represents around a quarter of total inorganic bromine trapped together with nitric acid on denuders (Legrand et al., 2016), we can conclude that, at least in East Antarctica, the contribution of HOBr in the chloride depletion of sea-salt aerosol cannot be totally ruled out, but it is certainly not a dominant process."*

*Concerning the role of bromide chemistry on nitric acid formation, though this is an interesting aspect, we feel that this point is out of the scope of this paper. At this point, we also would like to emphasize that indeed p-TOMCAT modelling including bromide chemistry suggests that at high latitude $BrONO_2$ photolysis is an important pathway for the production of $HNO_3$, but we have to consider that the Antarctic continent also experiences a major release of $NO_x$ from the snowpack, a process that is not considered by the model.*

P24, Figures 6/7: My second concern is again about the chlorine depletion in section 3.1. If the acidic replacement is indeed dominating chlorine release, them it could be possible to see a nearly 1:1 linear scatter relationship between chlorine depleted and $[H^+]$ derived. The $[H^+]$ can be derived from the two major acidic species (nitrate and sulphate without $SS\_SO_4$). Equally, a scatter plot between the ratio of r and ratio of $[H^+]/[Na^+]$ is also indicative to whether the replacement process is working. This examination can be made both for seasonal depletion data shown in figure 7 as well as for size-dependent depletion shown in figure 6. Instead of using $[H^+]$, a sum of $[NO_3]/M_{NO3}+2X[SO_4]/M_{SO4}$ can also be used in the scatter plot. here $M_{NO3}$ and $M_{SO4}$ are molecular weight of $NO_3$ and $SO_4$ respectively.

*We understand (and agree) what you means but Figure 7 was designed to discuss this point: may be it was not clear enough that in Figure 7 we expressed concentrations of chloride depletion and acidic components in molar units ("All concentrations are here expressed in molar units (in nEq m$^{-3}$)". We now specify this important point both in the figure caption and in the text. Furthermore, the relationship between chloride depletion and H$^+$ needs to be at least equal to 1 but can exceed one (and it is what is observed in late summer in Figure 7). This point is now emphasized before discussing the respective role of nitrate and sulfur species (that on a molar basis the sum of acidic species always exceed the chloride depletion):*
*"Figure 7 also indicates that, on a molar basis, the sum of acidic species always exceeds the chloride depletion. Whatever the time period, acidic sulfur particles are always present in large enough amounts to replace chloride in small sea-salt particles (Fig. 7a)."*

**Technical corrections:**

P2L32: add 'mass' between 'sodium' and 'ratios': *OK Done*
P8L10: remove 'to' before 'due to' *OK Done*
P8L20: 'Fig.6' should be 'Fig. 7'? *Sorry, in fact it is Figure 8 here. Corrected*

P8L18: 'larger' here should be 'smaller'? *Sorry, we refer to supermicron (not submicron) particles and larger is correct. Corrected.*
P9L19: remove the extra 'and' in 'or/and and too high' *OK Done*
P12L3. A reference is needed to 'high dust levels that neutralized $HNO_3$ and H2SO4. *OK, we add a reference (Usher et al., 2003 )*
P12L29: change 'part' to 'parts' *OK Done*

In most of figures, change NO3 to $NO_3^-$, SO4 to $SO_4^=$ , Na to $Na^+$. *OK Done.*

P22 figure 4: It would be good to add 'r' line to high light the 'relative depletion factor. *It is possible but we feel that this figure is already heavy and is mainly dedicated to show (1) the timing of the recovery of nitric acid in the gas phase versus the increase of nitrate on aerosol, (2) to compare calculated chloride depletion in HV and observed HCl in denuder tubes. Anyway r values are already show in Figure 2 (together with chloride depletion).*

P24 figure 6. Please add a line of total sulphate derived based on MAS (without SS_SO4). *It is possible but we feel that this figure is already heavy and is mainly dedicated to show on which particles nitrate is staying. Anyway examples of the size distribution of sulfate are shown in Figures 8 and 9.*

P25 figure 7b: is the number '0.64' in '(0.64-3.5 µm)' actually 0.46? why there is a gap between 0.46 to 0.64 µm ? same question in P6 L25. *It is an arbitrary cut but it show more clearly the difference between small and large particles.*

P28: what do the two dash lines represent for?
*OK we now specify in the figure caption: "The two horizontal dashed lines (in blue) refer to the R value in seawater (0.25, upper line) and in strongly fractionated sea-salt aerosol as observed in winter at the coast (0.07, lower line)."*

---

## Author Comment (AC2) · 16 Sep 2017

**Answer to Referee #3**

*We would like to first thank the reviewer for their comments which have emphasised to us that we needed to explain better some underlying concepts. We hope these additional explanations will answer most of the comments*

The manuscript presents multi-year measurement results of seal salt aerosols and associated ion composition as well as HCl and HNO3 acid gases in the central Antarctic site. With these data, the authors examine the chloride depletion relative to sodium with respect to freshly emitted sea salt aerosols and the sulfate depletion relative to sodium with respect to the composition of sea water. The seasonal variability of such depletion, the role of acidic sulfur aerosol and nitric acid in the depletion, and the contribution sea-ice and open ocean emissions to the sea salt aerosols load are investigated. The reported data are valuable and such kind of study should be a welcome addition to the literature on Antarctic environmental and atmospheric chemistry research. On the other hand, the manuscript appears to be not well written and some discussions are ambiguous without a clear clue to follow. Especially there is a lackness of in-depth analysis based on the fundamental chemistry. In my opinion the quality of this manuscript is not high enough to be published in ACP for its current version. Below are my comments in detail.

*We note that, in the quick report, our paper was evaluated as "good" for the aspect "scientific quality" and "data presentation".*

What does the authors mean in the term of "acidic sulfur aerosol" (e.g., in P1, L22)? The authors also define nitrate as "acidic compounds" (P6, L26-28). Do they refer to HNO3 in aerosols? Note that SO4= and NO3- should be taken as neutral, instead of acidic, ions in water solution. They calculated the acidic sulfur component as the sum of non-sea-salt sulfate plus MSA after subtracting the amount of ammonium (P6, L27- 28). What does the residual mean in acid-base equilibrium chemistry? The authors should have investigated the balance status between cations and anions in measured aerosols before analyzing the chlorine depletion by so called replacement reactions. Figure 7 and related discussions (P7, L10-17) provide some information. However, only when the full set of ions have been taken into account and, if possible, neural compounds (e.g., CaSO4 , Na2SO4, (NH4)2SO4, and NaNO3) diagnosed, the dis-placement process could be understood clearly. Note that there is a neutralization order for the ions in a solution, e.g., SO4= > NO3- for anions and Ca2+ > Na+ for cations. From Figure 7c, one cannot see whether the displacement had happened or not. Only by investigating NO3-, Na+ and other associated ions in the aerosols, the role of HNO3 in the displacement might be seen. Note that partitioning of HNO3 be tween gaseous and aerosol phases depends not only on the acidity of liquid aerosols but also atmospheric temperature.

*OK we understand that we made an assumption that readers are familiar with polar atmospheric chemistry and central Antarctic ice core chemistry. So before presenting the calculations of acidic sulfur we remedy that by presenting, as recommended by the reviewer, the ionic balance of aerosol at Concordia. This should indicate very clearly that, apart from the sea salt components, the main cation associated with nitrate and sulfate is H+ :*
*"An examination of the role over time of chemical species possibly involved in aerosol de-chlorination is reported in Fig. 7, the amount of chloride loss being compared on a molar basis to the main atmospheric acidic components. The mean ionic composition of aerosol collected in spring at Concordia is reported in Table 1. It shows that, apart from sea-salt components, aerosol present at Concordia in spring/summer consists of sulfate (1.2 nEq m⁻³), nitrate (0.08 nEq m⁻³), MSA (0.05 nEq m⁻³), and ammonium (0.18 nEq m⁻³). Other non-*

*sea-salt components (nssK⁺, nssCa²⁺, and oxalate) remain at the level of 0.01 nEq m⁻³ or less (Table 1). From that, we have calculated the acidic sulfur component as the sum of non-sea-salt sulfate plus MSA after subtracting the amount of ammonium."*

*Table 1. Mean chemical composition of aerosol collected at Concordia from October to December (2006-2015). Values in bold and in parenthesis refer to the non-sea-salt components of aerosol. All concentrations are expressed in nEq m⁻³.*

| $Na^+$ | $NH_4^+$ | $K^+$ | $Mg^{2+}$ | $Ca^{2+}$ | $Cl^-$ | $NO_3^-$ | $SO_4^{2-}$ | $CH_3SO_3^-$ | $C_2O_4^{2-}$ |
|---|---|---|---|---|---|---|---|---|---|
| 0.47 | 0.18 **(0.18)** | 0.02 **(0.01)** | 0 .11 | 0.02 **(<0.01)** | 0 .08 | 0.08 **(0.08)** | 1.25 **(1.19)** | 0.05 **(0.05)** | 0.015 **(0.015)** |

*Concerning your remark that "From Figure 7c, one cannot see whether the displacement had happened or not.":*
*Indeed, because only a small fraction of total nitrate is on sea-salt particles and figure 7c shows the gas phase fraction which (independently of sea-salt) increases from spring to summer with the recovery of the photochemistry and NOx emissions from the snowpack.*

For Section 3.2 of the manuscript, neither the terms of SO4, biogenic sulfate, ssSO4, nssSO4 and MSA and are well defined, nor their measuring (or calculating) methods clearly introduced.

*We think that the readers of Atmospheric Chemistry and Physics are already familiar with most of this terminology, and for example know that SO4 is sulphate. The method of calculation of nssSO4 was already defined on Page 2, line 16. The calculation of ssSO4 was already defined on Page 9, line 6. We think that the wording "biogenic sulphate" would be understood by the readers of ACP. We indeed missed spelling out MSA, and it is now done in the revised version in the introduction: "They also pointed out that, even when examining the chemical composition of particles deposited on the top stages of the impactor, there can be a significant underestimation of the degree of fractionation of sea-salt particles due to a residual presence of biogenic sulfate, as indicated by the presence of MSA (methanesulfonic acid). »*
*Adding this last sentence, we also clarify what is biogenic sulfate (see below).*
*Measurements are detailed in section 2, calculations are presented in this section. Please tell us if we need to do more but this seems sufficient for readers with any familiarity with the topic, and the justification for these calculations for those less familiar can easily be found in the cited references.*

It is difficult for me to follow the discussions as the assumption (e.g., the relationship between biogenic sulfate and MSA in P8, L22-25) has not been fully based one fundamental atmospheric chemistry and physics. There is a doubt that the regression results from small particles can be applied to large ones.

*We do not understand the argument for such a statement that, in any case, is not supported by observations. This point was already discussed in the text and we have included another sentence (mentioning that this has been previously discussed and published by Jourdain et al., 2007): "For the 5 lowest stages (smallest particle sizes) of the impactor, we find that the ratio of nssSO₄/MSA is reasonably constant (red dots in Fig 8e (or Fig 9e) falling on a straight line through zero). This then supports the assumption that in winter the size distributions of biogenic sulfate and MSA are the same and that the nssSO₄/MSA ratio is constant over the entire size distribution (see also Legrand et al., this*

*issue). The similarity of the nssSO$_4$/MSA ratio over the entire size distribution was already pointed out by Jourdain et al. (2007). «*

The authors refers to a companion paper: "Legrand, M., Preunkert, S., Weller, R., Zipf, L., Elsässer, C., Merchel, S., Rugel, G., and Wagenbach, D: Year-round (2006-2015) record of bulk and size-segregated aerosol composition in central Antarctica (Concordia site) Part 1: Sulfur derived aerosol (MSA and sulfate), this issue" (note that both companion papers are labelled as "Part 1").
*Thanks for identifying this typographic error.*

Considering that sulfate aerosol is a large and important part of this manuscript, I would suggest the paper not to be separated into two parts.
> *This suggestion is unrealistic. The topics of the two papers are totally different: part one is dedicated to the origin of sea-salt and its degree of fractionation over Antarctica, part 2 focuses on the understanding of the behaviour of biogenic sulfate and MSA.*
> *Part 1: is 17 pages of text (including references), 1 table, and 13 figures, Part 2: is 22 pages, 5 Tables, and 12 figures. Mixing the two papers (even considering the overlaps in section "sites, sampling and methods", and possible mixing 2 figures together), the resulting paper would be far too long and we feel this is not realistic.*

Sulfate depletion relative to sodium has been attributed to sea-ice related emissions due to precipitation of mirabilite (Na2SO4.10H2O) during freezing of seawater. As both sulfate and sodium will loss with the deposition of mirabilite, how the depletion of sulfate relative to sodium occurs should be explained quantitatively in detail.
> *This aspect is introduced and well referenced in the introduction (lines 10 to 21), and it is now a well recognized process. It is obvious that because there is much more sodium than sulfate in sea salt, precipitation of mirabilite removes proportionally more sulfate, leading to a depletion; we do not feel this needs to be explained again.*

The manuscript needs to be concise and formulas (1), (3), (4), (6), and (7) and associated descriptions can be moved to the Supplement.
> *Since theses calculations address two very different processes: equation 1 to 4 refer to calculations for summer (chloride depletion), whereas equations 5 to 7 to sulphate depletion in winter, they have to appear in the corresponding paragraph dedicated to summer and winter, respectively. Mixing them in a supplement would be more confusing, and we do not feel this would help the reader.*

The authors use the p-TOMCAT model to confirm their conclusion on the source of sea salt aerosols (P10, L11-23). But the model and its simulation results have not been well evaluated and introduced in this manuscript. These discussions (P10, L11-23) provide no more convincing information than the backward trajectory analysis described in previous paragraph.
> *As discussed in these lines, these model simulations were evaluated against observations in the cited previous papers. We feel that these previous works have to be reported here since the present paper clearly assesses (on a far more robust basis) previous observations. The advantage of the p-TOMCAT simulations compared to back trajectories is that they quantitatively assess the source of sea salt aerosol while the back trajectories merely assess the source of air masses.*

I do not think that Sect 3.3 should be included in this manuscript as it appears to provide no strong support to explain the observational results of this study.

*We disagree. The ice core data can, in no way, here support the atmospheric observations discussed in this study simply because, as clearly mentioned, post depositional effects modified the original atmospheric signal. On the contrary, as clearly introduced in the paper, the atmospheric studies described in this paper are needed to better understand ice core signals. The reviewer 1 clearly highlighted this point : "In general, this is a novel and deepening study of aerosol depletions for chlorine and sulphate at a central Antarctica site. It is well written and with huge scientific implications for our understanding of the sea salt ice records taken from inland Antarctica."*

*Also the title of this paragraph is very explicit: "Implications for ice core studies" and not the reverse. From our experience some previous papers also ended with a discussion on "implications for ice core": see for instance the JGR paper "Preunkert, S., Jourdain, B., Legrand, M., Udisti, R., Becagli, S., and Cerri, O.: Seasonality of sulfur species (dimethyl sulfide, sulfate, and methanesulfonate) in Antarctica: Inland versus coastal regions, J. Geophys. Res., 113, D15302, doi:10.1029/2008JD009937, 2008.*

---

## Author Comment (AC3) · 16 Sep 2017

The authors address a pressing topic; sea salt, its size distribution in aerosols and co-located gas phase measurements over annual cycles. The interpretation of sea salt records in Antarctica is relevant to sea ice distribution patterns recorded in glacial records. This manuscript contributes significantly to this discussion, demonstrating that approximately half the sea salt aerosol load comes from open ocean versus sea ice. The relationship of sea salt, aerosol acidification and the production of HCl and HNO3 is also examined by considering denuder measurements of acid gases and their effect on aerosol characteristics. The authors argued that HCl emission from surface snow or differences in the lifetime of HCl versus sea salt transported from the coast drove differences observed between denuder and aerosol measurements. These results are novel and intriguing. This paper provides constraints to some long-standing questions regarding the source of anomalies in the ratio of Na to Cl in snow, whether and by how much Cl in aerosols is depleted in the continental Antarctic and in ice and the factors driving it. These points are all strong and convincing. Arguments about sea salt and biogenic sulfate however, are less well-rounded: likely this is because detail is provided in a separate manuscript dealing with sulfate and sulfate to sodium ratios. The manuscript is strong, but some sections, particularly with respect to sulfate and sulfate to sodium ratios need additional attention prior to publication.

Portions of this manuscript associated with interpretation of sulfate needs additional description (that perhaps can be found in the accompanying paper on sulfur-derived aerosol) to be fully understandable. Page 8 line 24 refers to quantifying biogenic sulfate using MSA and the relationship between MSA and sulfate on particles with a diameter <0.46 microns. This assumes that all non sea salt sulfate in aerosols <0.46 microns in diameter is biogenic. Is this because anthropogenic sulfate is negligible and volcanic sulfate is absent in the Antarctic interior? A previous study in the Arctic demonstrated that MSA in for aerosols <0.49 microns correlated with the presence of fine particles rather than with biogenic sulfate (Rempillo et al., 2010). In that study the surface of <0.49 micron aerosol, rather than biogenic sulfate, were the dominant factor affecting MSA formation. The authors need to demonstrate why anthropogenic and volcanic sulfate are not factors in MSA formation at this continental Antarctic site. Line 23 on page 8 ".. we therefore have corrected the concentrations of sulfate present on the upper stages of the impactor for the (nss rather than biogenic contribution)".

> *Thank you for this comment: Yes, at these high southern latitudes, anthropogenic and volcanic sources can be neglected with respect to DMS oxidation. Such a statement was drawn from model approaches (Gondwe et al., 2003), sulfur isotopic measurements (Patris et al., 2000) and examination of MSA to nssSO4 (Minikin et al., 1998). Furthermore, indeed this point is discussed in the accompanying paper. It is concluded that, even for the low winter levels of nssSO4 (6 ng m$^{-3}$), its presence is mainly attributed to long-range transport of marine biogenic emissions from mid-latitude, the non-biogenic sulfate level remaining limited to 1 ng m$^{-3}$. But we have considered your comment and in section 3.2., we first refer to sea-salt sulphate and non-sea-salt sulphate modes, then we refer to the accompanying paper arguing that we can assume that non-sea-salt sulfate present in winter there mainly originates from marine biogenic emissions, and finally use the wording "biogenic sulfate".*

*Gondwe, M., Krol, M., Gieskes, W., Klaassen, W., and de Baar, H.: The contribution of ocean-leaving DMS to the global atmospheric burdens of DMS, MSA, SO$_2$, and NSS SO$_4$, Global Biogeochem. Cycles, 17(2), 1056, doi:10.1029/2002GB001937, 2003.*

*Minikin, A., Legrand, M., Hall, J., Wagenbach, D., Kleefeld, C., Wolff, E., Pasteur, E. C., and Ducroz, F.: Sulfur-containing spacies (sulfate and methanesulfonate) in coastal Antarctic aerosol and precipitation, J. Geophys. Res., 103, 10,975-10,990, 1998.*

*Patris, N., Delmas, R. J., and Jouzel, J.: Isotopic signatures of sulfur in shallow Antarctic ice cores, J. Geophys. Res., 105(D6), 7071-7078, doi:10.1029/1999JD900974.*

On page 10 how were the values of 40% and 60% of travel time within the boundary layer determined? This is not evident from the examples of air mass back trajectories provided in Figure 11. Is there a reference or data to support a boundary layer of ~600 m for open ocean and over sea-ice in the Antarctic at this time of year? The discussion suggests a quantified analysis that is not shared with the reader which makes this section less convincing.

*You are right, our wording concerning 600 m was confusing. Also Figure 11 only shows the location of the trajectories and not the altitude. The altitude of the trajectories is provided by trajectories calculations but not shown here. So we reworded the paragraph as follows:*

*"The temporal variability of sea-salt aerosol reaching Dome C, its level and composition with respect to the sulfate relative to sodium fractionation, were examined in the light of air masses reaching the site, as shown by calculated 10-day backward trajectories. The model was run every 6 h in backward mode for three different altitudes (0, 250, and 500 m above ground level, agl) for time periods corresponding to impactor run sampling. It is seen that when air masses arriving at Dome C have spent more than two days (up to 4 days) over oceanic sectors their mean sodium concentrations reach 47 ± 21 ng m$^{-3}$ against 12 ± 5 ng m$^{-3}$ when air masses have spent less than 0.5 days over oceans. Some typical examples of 10-day backward trajectories corresponding to strong, mid and weak sulfate relative to sodium fractionations are reported in Fig. 11. Backward air mass trajectory calculations also document the altitude of trajectories (not shown) and we calculated the time that air masses reaching Dome C were traveling below 600 m elevation over oceans, distinguishing between sea-ice and open ocean areas. The elevation of 600 m asl corresponds to the middle of the boundary layer whose the thickness over oceanic regions is typically 1000-1500 m. Selecting the impactor runs corresponding to sampling time periods over which the 10-day backward trajectories indicate more than one day of travel over the ocean, we find an overall decreasing trend of R (i.e. a stronger sulfate depletion relative to sodium) when air masses arriving at Dome C (0 m agl) have a longer contact with sea-ice than with open-ocean (R$^2$ = 0.4). For example, from 9 to 23 September 2006, air masses reaching Concordia had travelled more than two days over oceanic sectors, with 40% and 60% of travel time below 600 m elevation over sea-ice and open-ocean boundary, respectively. The corresponding R value is low (R = 0.1, Fig. 11a), consistently suggesting a very significant contribution of sea-ice-related processes to the sodium level. At the opposite, from 28 August to 3 September 2008 (Fig. 11d), air masses arriving at Dome C have only travelled 5% of time below 600 m elevation over sea-ice (95% over open-ocean). Intermediate situations, from 26 July to 9 August 2007 (Fig. 11b) or 28*

*August to 11 September 2009 (Fig. 11c) are characterized by R values close to 0.15 and 20-25 % of travel time below 600 m elevation over sea-ice (75-80% over open-ocean)."*

Line 30 on page 10 is confusing "A slight increase of the sulfate depletion in particles smaller than higher than one micron. . .."

*We agree: this sentence is confusing and add nothing with respect to the preceding sentence. So we removed it.*

---

## Author Comment (AC4) · 16 Sep 2017

Dear Dr. Legrand,

My apologize for such a long time for the open discussions of your manuscript. Your manuscript did undergo an unusual experience, which is certainly unpleasant for you as well as for us. Two referees had accepted our invitation to review your manuscript in April. But both of them failed to submit their reports in early July when the open discussions should be closed normally for your manuscript. It took a few weeks more for me to contact them and to nominate other potential referees. Although I found another two referees agreeing to review your manuscript, unfortunately the report from one referee was missing again by the initial and extended deadlines. While promising to submit the review report soon, one referee said in his/her personal email to me that the paper is good but tough to get through and it's taken longer than expected. Actually, I share the same feeling with that referee when reading your manuscript.

Now we have gotten two review reports. While both referees admire import value of your data and significance of your work, one of them rates a low value of the quality of your manuscript especially in presentation. I agree with the referee (Referee 3) in that the manuscript needs to be focused more on the analysis of chemical processes. Actually, another referee (Referee 1) also suggested that additional chemical process be considered for chlorine depletion relative to sodium with respect to freshly emitted sea salt aerosols.

I noted that the sulfate aerosol issue has been intensively addressed by a companion paper of this manuscript (Legrand et al., 2017), which was also published in ACPD. Therefore, you may refer to that paper for the filtering of biogenic sulfate aerosols and, as suggested by the referee, focus more on the ionic chemistry involved in sulfate depletion relative to sodium with respect to the composition of sea water.

I also agree with the referee in that the discussions on implications for ice core stud- ies (Sect. 3.3) should be skipped over if these discussion help little to explain your measurement and analysis results presented in the preceding sections.
In summary, I think that your manuscript needs substantial revisions based on the comments from the referees. You are welcome to submit the revised manuscript if you think that all the issues they raised can be well addressed. Your manuscript will be sent to the referees for further review, and the final decision can be made then.
If you have any questions, please do not hesitate to contact me. C2
Sincerely,
Jianzhong Ma

Reference: Legrand, M., Preunkert, S., Weller, R., Zipf, L., Elsässer, C., Merchel, S., Rugel, G., and Wagenbach, D.: Year-round record of bulk and size-segregated aerosol composition in central Antarctica (Concordia site) Part 2: Biogenic sulfur (sulfate and methanesulfonate) aerosol, Atmos. Chem. Phys. Discuss., 2017, 1-39, 10.5194/acp- 2017-305, 2017.

**Answers to the Editor**

Thank you very much for your comments. We have now three reviewers and two of them are very positive, both considering the paper as well written and important for atmospheric chemists as well as the ice core community (reviewer 1: "It is well written and with huge scientific implications for our understanding of the sea salt ice records taken from inland Antarctica."), reviewer 4 (for instance "This paper provides constraints to some long-standing questions regarding the source of anomalies in the ratio of Na to Cl in snow, whether and by how much Cl in aerosols is depleted in the continental Antarctic and in ice and the factors driving it."). In our response, we carefully considered all points raised by reviewer 1 and reviewer 4.

Concerning reviewer 3, we have carefully addressed the questions he raised on the poor discussion we had of the ionic balance and have now reported the mean ionic balance of aerosol (Table 1).

We cannot, however, follow two of the proposed changes.

First, it is clearly not realistic to mix the two manuscripts. Indeed in our response the reviewer 3 we argued that: "*The topics of the two papers are totally different: part one is dedicated to the origin of sea-salt and its degree of fractionation over Antarctica, part 2 focuses on the understanding of the behaviour of biogenic sulfate and MSA.*

*Part 1: is 17 pages of text (including references), 1 table, and 13 figures, Part 2: is 22 pages, 5 Tables, and 12 figures. Mixing the two papers (even considering the overlaps in section*

*"sites, sampling and methods", and possible mixing 2 figures together), the resulting paper would be far too long and we feel this is not realistic."*

Second, we don't agree with the suggestion to remove the section on ice core implications. Indeed, as argued in our answer: *"The ice core data can, in no way, here support the atmospheric observations discussed in this study simply because, as clearly mentioned, post depositional effects modified the original atmospheric signal. On the contrary, as clearly introduced in the paper, the atmospheric studies described in this paper are needed to better understand ice core signals. The reviewer 1 clearly highlighted this point: "In general, this is a novel and deepening study of aerosol depletions for chlorine and sulphate at a central Antarctica site. It is well written and with huge scientific implications for our understanding of the sea salt ice records taken from inland Antarctica."*
*Also the title of this paragraph is very explicit: "Implications for ice core studies" and not the reverse. From our experience some previous papers also ended with a discussion on "implications for ice core": see for instance the JGR paper "Preunkert, S., Jourdain, B., Legrand, M., Udisti, R., Becagli, S., and Cerri, O.: Seasonality of sulfur species (dimethyl sulfide, sulfate, and methanesulfonate) in Antarctica: Inland versus coastal regions, J. Geophys. Res., 113, D15302, doi:10.1029/2008JD009937, 2008. »*

---

## Referee Report (RR1)

The revised manuscript has carefully addressed my questions raised in the 1st round comment. Basically, I am happy with their responses, even though they did go further on the issue regarding the role of bromine on $HNO_3$ formation (e.g. via BrONO2 hydrolysis reaction). Anyway, I agree with their argument that this issue is slightly out of the scope of this current paper.

In general, the revised paper has made some changes according to all the three reviewers comments, e.g. with a new table being added to address the issue raised by reviewer #3 on the ionic balance of aerosols at Concordia, which is very helpful. The added texts and discussions regarding, for example, the HOBr effect on the chlorine depletion, the anthropogenic and volcanic contribution to sulphate formation in polar regions, as well as the back-trajectory results, all well improve the quality of the manuscript and strengthen their conclusions derived. Thus, as I concluded in my first round review, I again recommend this manuscript to be accepted for publication after just a few technical corrections (as shown below).

Technical corrections:

In the text, both 'asl' and 'agl' are used. Would it be better to stick on one term? Also the 'asl' (at its first appearance on P3L30) was not even defined; while the 'agl' was defined twice (at P4L33 and P10L11).

I noticed that in most figures sub-/super-scripts are not properly applied for chemical species and ions. For example, in Fig. 4d, Na+ is used instead of $Na^+$; NO3- is used instead of to $NO_3^-$. Similar issues are also spotted in fig. 3, 5 and 6. In particular, in fig. 8 and 9, SO4- - is used, which looks quite strange. All these should be corrected before publication.